# Reward-based training of recurrent neural networks for cognitive and value-based tasks

**H Francis Song[1], Guangyu R Yang[1], Xiao-Jing Wang[1,2]\***

[1]Center for Neural Science, New York University, New York, United States; [2]NYU-ECNU Institute of Brain and Cognitive Science, NYU Shanghai, Shanghai, China

**Abstract** Trained neural network models, which exhibit features of neural activity recorded from behaving animals, may provide insights into the circuit mechanisms of cognitive functions through systematic analysis of network activity and connectivity. However, in contrast to the graded error signals commonly used to train networks through supervised learning, animals learn from reward feedback on definite actions through reinforcement learning. Reward maximization is particularly relevant when optimal behavior depends on an animal's internal judgment of confidence or subjective preferences. Here, we implement reward-based training of recurrent neural networks in which a value network guides learning by using the activity of the decision network to predict future reward. We show that such models capture behavioral and electrophysiological findings from well-known experimental paradigms. Our work provides a unified framework for investigating diverse cognitive and value-based computations, and predicts a role for value representation that is essential for learning, but not executing, a task.

**\*For correspondence:** xjwang@nyu.edu

**Competing interests:** The authors declare that no competing interests exist.

## Introduction

A major challenge in uncovering the neural mechanisms underlying complex behavior is our incomplete access to relevant circuits in the brain. Recent work has shown that model neural networks optimized for a wide range of tasks, including visual object recognition (*Cadieu et al., 2014*; *Yamins et al., 2014*; *Hong et al., 2016*), perceptual decision-making and working memory (*Mante et al., 2013*; *Barak et al., 2013*; *Carnevale et al., 2015*; *Song et al., 2016*; *Miconi, 2016*), timing and sequence generation (*Laje and Buonomano, 2013*; *Rajan et al., 2015*), and motor reach (*Hennequin et al., 2014*; *Sussillo et al., 2015*), can reproduce important features of neural activity recorded in numerous cortical areas of behaving animals. The analysis of such circuits, whose activity and connectivity are fully known, has therefore re-emerged as a promising tool for understanding neural computation (*Zipser and Andersen, 1988*; *Sussillo, 2014*; *Gao and Ganguli, 2015*). Constraining network training with tasks for which detailed neural recordings are available may also provide insights into the principles that govern learning in biological circuits (*Sussillo et al., 2015*; *Song et al., 2016*; *Brea and Gerstner, 2016*).

Previous applications of this approach to 'cognitive-type' behavior such as perceptual decision-making and working memory have focused on supervised learning from graded error signals. Animals, however, learn to perform specific tasks from reward feedback provided by the experimentalist in response to definite actions, i.e., through reinforcement learning (*Sutton and Barto, 1998*). Unlike in supervised learning where the network is given the correct response on each trial in the form of a continuous target output to be followed, reinforcement learning provides evaluative feedback to the network on whether each selected action was 'good' or 'bad.' This form of feedback allows for a graded notion of behavioral correctness that is distinct from the graded difference

**eLife digest** A major goal in neuroscience is to understand the relationship between an animal's behavior and how this is encoded in the brain. Therefore, a typical experiment involves training an animal to perform a task and recording the activity of its neurons – brain cells – while the animal carries out the task. To complement these experimental results, researchers "train" artificial neural networks – simplified mathematical models of the brain that consist of simple neuron-like units – to simulate the same tasks on a computer. Unlike real brains, artificial neural networks provide complete access to the "neural circuits" responsible for a behavior, offering a way to study and manipulate the behavior in the circuit.

One open issue about this approach has been the way in which the artificial networks are trained. In a process known as reinforcement learning, animals learn from rewards (such as juice) that they receive when they choose actions that lead to the successful completion of a task. By contrast, the artificial networks are explicitly told the correct action. In addition to differing from how animals learn, this limits the types of behavior that can be studied using artificial neural networks.

Recent advances in the field of machine learning that combine reinforcement learning with artificial neural networks have now allowed Song et al. to train artificial networks to perform tasks in a way that mimics the way that animals learn. The networks consisted of two parts: a "decision network" that uses sensory information to select actions that lead to the greatest reward, and a "value network" that predicts how rewarding an action will be. Song et al. found that the resulting artificial "brain activity" closely resembled the activity found in the brains of animals, confirming that this method of training artificial neural networks may be a useful tool for neuroscientists who study the relationship between brains and behavior.

The training method explored by Song et al. represents only one step forward in developing artificial neural networks that resemble the real brain. In particular, neural networks modify connections between units in a vastly different way to the methods used by biological brains to alter the connections between neurons. Future work will be needed to bridge this gap.

between the network's output and the target output in supervised learning. For the purposes of using model networks to generate hypotheses about neural mechanisms, this is particularly relevant in tasks where the optimal behavior depends on an animal's internal state or subjective preferences. In a perceptual decision-making task with postdecision wagering, for example, on a random half of the trials the animal can opt for a sure choice that results in a small (compared to the correct choice) but certain reward (*Kiani and Shadlen, 2009*). The optimal decision regarding whether or not to select the sure choice depends not only on the task condition, such as the proportion of coherently moving dots, but also on the animal's own confidence in its decision *during the trial*. Learning to make this judgment cannot be reduced to reproducing a predetermined target output without providing the full probabilistic solution to the network. It can be learned in a natural, ethologically relevant way, however, by choosing the actions that result in greatest overall reward; through training, the network learns from the reward contingencies alone to condition its output on its internal estimate of the probability that its answer is correct.

Meanwhile, supervised learning is often not appropriate for value-based, or economic, decision-making where the 'correct' judgment depends explicitly on rewards associated with different actions, even for identical sensory inputs (*Padoa-Schioppa and Assad, 2006*). Although such tasks can be transformed into a perceptual decision-making task by providing the associated rewards as inputs, this sheds little light on how value-based decision-making is learned by the animal because it conflates external with 'internal,' learned inputs. More fundamentally, reward plays a central role in all types of animal learning (*Sugrue et al., 2005*). Explicitly incorporating reward into network training is therefore a necessary step toward elucidating the biological substrates of learning, in particular reward-dependent synaptic plasticity (*Seung, 2003*; *Soltani et al., 2006*; *Izhikevich, 2007*; *Urbanczik and Senn, 2009*; *Frémaux et al., 2010*; *Soltani and Wang, 2010*; *Hoerzer et al., 2014*; *Brosch et al., 2015*; *Friedrich and Lengyel, 2016*) and the role of different brain structures in learning (*Frank and Claus, 2006*).

In this work, we build on advances in recurrent policy gradient reinforcement learning, specifically the application of the REINFORCE algorithm (*Williams, 1992*; *Baird and Moore, 1999*; *Sutton et al., 2000*; *Baxter and Bartlett, 2001*; *Peters and Schaal, 2008*) to recurrent neural networks (RNNs) (*Wierstra et al., 2009*), to demonstrate reward-based training of RNNs for several well-known experimental paradigms in systems neuroscience. The networks consist of two modules in an 'actor-critic' architecture (*Barto et al., 1983*; *Grondman et al., 2012*), in which a decision network uses inputs provided by the environment to select actions that maximize reward, while a value network uses the selected actions and activity of the decision network to predict future reward and guide learning. We first present networks trained for tasks that have been studied previously using various forms of supervised learning (*Mante et al., 2013*; *Barak et al., 2013*; *Song et al., 2016*); they are characterized by 'simple' input-output mappings in which the correct response for each trial depends only on the task condition, and include perceptual decision-making, context-dependent integration, multisensory integration, and parametric working memory tasks. We then show results for tasks in which the optimal behavior depends on the animal's internal judgment of confidence or subjective preferences, specifically a perceptual decision-making task with postdecision wagering (*Kiani and Shadlen, 2009*) and a value-based economic choice task (*Padoa-Schioppa and Assad, 2006*). Interestingly, unlike for the other tasks where we focus on comparing the activity of units in the decision network to neural recordings in the dorsolateral prefrontal and posterior parietal cortex of animals performing the same tasks, for the economic choice task we show that the activity of the value network exhibits a striking resemblance to neural recordings from the orbitofrontal cortex (OFC), which has long been implicated in the representation of reward-related signals (*Wallis, 2007*).

An interesting feature of our REINFORCE-based model is that a reward baseline—in this case, the output of a recurrently connected value network (*Wierstra et al., 2009*)—is essential for learning, but not for executing the task, because the latter depends only on the decision network. Importantly, learning can sometimes still occur without the value network but is much more unreliable. It is sometimes observed in experiments that reward-modulated structures in the brain such as the basal ganglia or OFC are necessary for learning or adapting to a changing environment, but not for executing a previously learned skill (*Turner and Desmurget, 2010*; *Schoenbaum et al., 2011*; *Stalnaker et al., 2015*). This suggests that one possible role for such circuits may be representing an accurate baseline to guide learning. Moreover, since confidence is closely related to expected reward in many cognitive tasks, the explicit computation of expected reward by the value network provides a concrete, learning-based rationale for confidence estimation as a ubiquitous component of decision-making (*Kepecs et al., 2008*; *Wei and Wang, 2015*), even when it is not strictly required for performing the task.

Conceptually, the formulation of behavioral tasks in the language of reinforcement learning presented here is closely related to the solution of partially observable Markov decision processes (POMDPs) (*Kaelbling et al., 1998*) using either model-based belief states (*Rao, 2010*) or model-free working memory (*Todd et al., 2008*). Indeed, as in *Dayan and Daw, (2008)* one of the goals of this work is to unify related computations into a common language that is applicable to a wide range of tasks in systems neuroscience. Such policies can also be compared more directly to behaviorally 'optimal' solutions when they are known, for instance to the signal detection theory account of perceptual decision-making (*Gold and Shadlen, 2007*). Thus, in addition to expanding the range of tasks and neural mechanisms that can be studied with trained RNNs, our work provides a convenient framework for the study of cognitive and value-based computations in the brain, which have often been viewed from distinct perspectives but in fact arise from the same reinforcement learning paradigm.

## Results

### Policy gradient reinforcement learning for behavioral tasks

For concreteness, we illustrate the following in the context of a simplified perceptual decision-making task based on the random dots motion (RDM) discrimination task as described in *Kiani et al. (2008)* (*Figure 1A*). In its simplest form, in an RDM task the monkey must maintain fixation until a 'go' cue instructs the monkey to indicate its decision regarding the direction of coherently moving

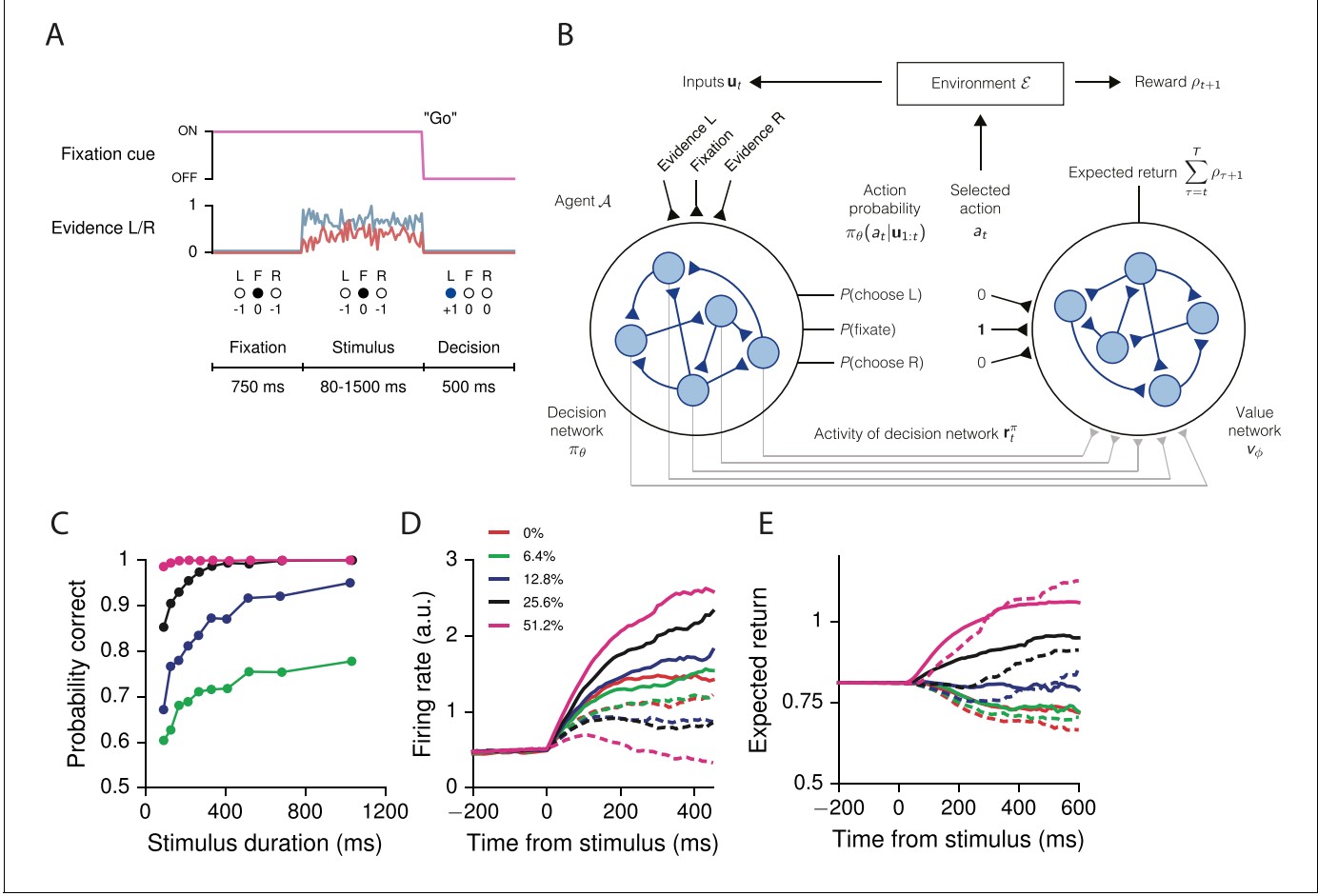

**Figure 1.** Recurrent neural networks for reinforcement learning. (**A**) Task structure for a simple perceptual decision-making task with variable stimulus duration. The agent must maintain fixation ($a_t = \mathrm{F}$) until the go cue, which indicates the start of a decision period during which choosing the correct response ($a_t = \mathrm{L}$ or $a_t = \mathrm{R}$) results in a positive reward. The agent receives zero reward for responding incorrectly, while breaking fixation early results in an aborted trial and negative reward. (**B**) At each time $t$ the agent selects action $a_t$ according to the output of the decision network $\pi_\theta$, which implements a policy that can depend on all past and current inputs $\mathbf{u}_{1:t}$ provided by the environment. In response, the environment transitions to a new state and provides reward $\rho_{t+1}$ to the agent. The value network $v_\phi$ uses the selected action and the activity of the decision network $\mathbf{r}_t^\pi$ to predict future rewards. All the weights shown are plastic, i.e., trained by gradient descent. (**C**) Performance of the network trained for the task in (**A**), showing the percent correct by stimulus duration, for different coherences (the difference in strength of evidence for L and R). (**D**) Neural activity of an example decision network unit, sorted by coherence and aligned to the time of stimulus onset. Solid lines are for positive coherence, dashed for negative coherence. (**E**) Output of the value network (expected return) aligned to stimulus onset. Expected return is computed by performing an 'absolute value'-like operation on the accumulated evidence.

The following figure supplements are available for figure 1:

**Figure supplement 1.** Learning curves for the simple perceptual decision-making task.

**Figure supplement 2.** Reaction-time version of the perceptual decision-making task, in which the go cue coincides with the onset of stimulus, allowing the agent to choose when to respond.

**Figure supplement 3.** Learning curves for the reaction-time version of the simple perceptual decision-making task.

**Figure supplement 4.** Learning curves for the simple perceptual decision-making task with a linear readout of the decision network as the baseline.

dots on the screen. Thus the three possible actions available to the monkey at any given time are fixate, choose left, or choose right. The true direction of motion, which can be considered a *state* of the environment, is not known to the monkey with certainty, i.e., is *partially observable*. The monkey must therefore use the noisy sensory evidence to infer the direction in order to select the correct response at the end of the trial. Breaking fixation early results in a negative reward in the form of a timeout, while giving the correct response after the fixation cue is extinguished results in a positive reward in the form of juice. Typically, there is neither a timeout nor juice for an incorrect response during the decision period, corresponding to a 'neutral' reward of zero. The goal of this section is to give a general description of such tasks and how an RNN can learn a behavioral *policy* for choosing actions at each time to maximize its cumulative reward.

Consider a typical interaction between an experimentalist and animal, which we more generally call the environment $\mathcal{E}$ and agent $\mathcal{A}$, respectively (**Figure 1B**). At each time $t$ the agent chooses to perform actions $\mathbf{a}_t$ after observing inputs $\mathbf{u}_t$ provided by the environment, and the probability of choosing actions $\mathbf{a}_t$ is given by the agent's policy $\pi_\theta(\mathbf{a}_t|\mathbf{u}_{1:t})$ with parameters $\theta$. Here the policy is implemented as the output of an RNN, so that $\theta$ comprises the connection weights, biases, and initial state of the decision network. The policy at time $t$ can depend on all past and current inputs $\mathbf{u}_{1:t} = (\mathbf{u}_1, \mathbf{u}_2, \ldots, \mathbf{u}_t)$, allowing the agent to integrate sensory evidence or use working memory to perform the task. The exception is at $t = 0$, when the agent has yet to interact with the environment and selects its actions 'spontaneously' according to $\pi_\theta(\mathbf{a}_0)$. We note that, if the inputs give exact information about the environmental state $\mathbf{s}_t$, i.e., if $\mathbf{u}_t = \mathbf{s}_t$, then the environment can be described by a Markov decision process. In general, however, the inputs only provide partial information about the environmental states, requiring the network to accumulate evidence over time to determine the state of the world. In this work we only consider cases where the agent chooses one out of $N_a$ possible actions at each time, so that $\pi_\theta(\mathbf{a}_t|\mathbf{u}_{1:t})$ for each $t$ is a discrete, normalized probability distribution over the possible actions $a_1, \ldots, a_{N_a}$. More generally, $\mathbf{a}_t$ can implement several distinct actions or even continuous actions by representing, for example, the means of Gaussian distributions (**Peters and Schaal, 2008**; **Wierstra et al., 2009**). After each set of actions by the agent at time $t$ the environment provides a reward (or special observable) $\rho_{t+1}$ at time $t+1$, which the agent attempts to maximize in the sense described below.

In the case of the example RDM task above (**Figure 1A**), the environment provides (and the agent receives) as inputs a fixation cue and noisy evidence for two choices L(eft) and R(ight) during a variable-length stimulus presentation period. The strength of evidence, or the difference between the evidence for L and R, is called the coherence, and in the actual RDM experiment corresponds to the percentage of dots moving coherently in one direction on the screen. The agent chooses to perform one of $N_a = 3$ actions at each time: fixate ($a_t = \mathrm{F}$), choose L ($a_t = \mathrm{L}$), or choose R ($a_t = \mathrm{R}$). Here, the agent must choose F as long as the fixation cue is on, and then, when the fixation cue is turned off to indicate that the agent should make a decision, correctly choose L or R depending on the sensory evidence. Indeed, for all tasks in this work we required that the network 'make a decision' (i.e., break fixation to indicate a choice at the appropriate time) on at least 99% of the trials, whether the response was correct or not. A trial ends when the agent chooses L or R regardless of the task epoch: breaking fixation early before the go cue results in an aborted trial and a negative reward $\rho_t = -1$, while a correct decision is rewarded with $\rho_t = +1$. Making the wrong decision results in no reward, $\rho_t = 0$. For the zero-coherence condition the agent is rewarded randomly on half the trials regardless of its choice. Otherwise the reward is always $\rho_t = 0$.

Formally, a trial proceeds as follows. At time $t = 0$, the environment is in state $\mathbf{s}_0$ with probability $\mathcal{E}(\mathbf{s}_0)$. The state $\mathbf{s}_0$ can be considered the starting time (i.e., $t = 0$) and 'task condition,' which in the RDM example consists of the direction of motion of the dots (i.e., whether the correct response is L or R) and the coherence of the dots (the difference between evidence for L and R). The time component of the state, which is updated at each step, allows the environment to present different inputs to the agent depending on the task epoch. The true state $\mathbf{s}_0$ (such as the direction of the dots) is only partially observable to the agent, so that the agent must instead infer the state through inputs $\mathbf{u}_t$ provided by the environment during the course of the trial. As noted previously, the agent initially chooses actions $\mathbf{a}_0$ with probability $\pi_\theta(\mathbf{a}_0)$. The networks in this work almost always begin by choosing F, or fixation.

At time $t = 1$, the environment, depending on its previous state $\mathbf{s}_0$ and the agent's action $\mathbf{a}_0$, transitions to state $\mathbf{s}_1$ with probability $\mathcal{E}(\mathbf{s}_1|\mathbf{s}_0, \mathbf{a}_0)$ and generates reward $\rho_1$. In the perceptual decision-

making example, only the time advances since the trial condition remains constant throughout. From this state the environment generates observable $\mathbf{u}_1$ with a distribution given by $\mathcal{E}(\mathbf{u}_1|\mathbf{s}_1)$. If $t = 1$ were in the stimulus presentation period, for example, $\mathbf{u}_1$ would provide noisy evidence for L or R, as well as the fixation cue. In response, the agent, depending on the inputs $\mathbf{u}_1$ it receives from the environment, chooses actions $\mathbf{a}_1$ with probability $\pi_\theta(\mathbf{a}_1|\mathbf{u}_{1:1}) = \pi_\theta(\mathbf{a}_1|\mathbf{u}_1)$. The environment, depending on its previous states $\mathbf{s}_{0:1} = (\mathbf{s}_0, \mathbf{s}_1)$ and the agent's previous actions $a_{0:1} = (\mathbf{a}_0, \mathbf{a}_1)$, then transitions to state $\mathbf{s}_2$ with probability $\mathcal{E}(\mathbf{s}_2|\mathbf{s}_{0:1}, \mathbf{a}_{0:1})$ and generates reward $\rho_2$. These steps are repeated until the end of the trial at time $T$. Trials can terminate at different times (e.g., for breaking fixation early or because of variable stimulus durations), so that $T$ in the following represents the maximum length of a trial. In order to emphasize that rewards follow actions, we adopt the convention in which the agent performs actions at $t = 0, \ldots, T$ and receives rewards at $t = 1, \ldots, T+1$.

The goal of the agent is to maximize the sum of expected future rewards at time $t = 0$, or expected return

$$J(\theta) = \mathbb{E}_H\left[\sum_{t=0}^{T} \rho_{t+1}\right], \tag{1}$$

where the expectation $\mathbb{E}_H$ is taken over all possible trial histories $H = (\mathbf{s}_{0:T+1}, \mathbf{u}_{1:T}, \mathbf{a}_{0:T})$ consisting of the states of the environment, the inputs given to the agent, and the actions of the agent. In practice, the expectation value in *Equation 1* is estimated by performing $N_{\text{trials}}$ trials for each policy update, i.e., with a Monte Carlo approximation. The expected return depends on the policy and hence parameters $\theta$, and we use Adam stochastic gradient descent (SGD) (*Kingma and Ba, 2015*) with gradient clipping (*Graves, 2013*; *Pascanu et al., 2013b*) to find the parameters that maximize this reward (Materials and methods).

More specifically, after every $N_{\text{trials}}$ trials the decision network uses gradient descent to update its parameters in a direction that minimizes an objective function $\mathcal{L}^\pi$ of the form

$$\mathcal{L}^\pi(\theta) = \frac{1}{N_{\text{trials}}} \sum_{n=1}^{N_{\text{trials}}} \left[-J_n(\theta) + \Omega_n^\pi(\theta)\right] \tag{2}$$

with respect to the connection weights, biases, and initial state of the decision network, which we collectively denote as $\theta$. Here $\Omega_n^\pi(\theta)$ can contain any regularization terms for the decision network, for instance an entropy term to control the degree of exploration (*Xu et al., 2015*). The key gradient $\nabla_\theta J_n(\theta)$ is given for each trial $n$ by the REINFORCE algorithm (*Williams, 1992*; *Baird and Moore, 1999*; *Sutton et al., 2000*; *Baxter and Bartlett, 2001*; *Peters and Schaal, 2008*; *Wierstra et al., 2009*) as

$$\nabla_\theta J_n(\theta) = \sum_{t=0}^{T} [\nabla_\theta \log \pi_\theta(\mathbf{a}_t|\mathbf{u}_{1:t})]\left[\sum_{\tau=t}^{T} \rho_{\tau+1} - v_\phi\left(\mathbf{a}_{1:t}, \mathbf{r}_{1:t}^\pi\right)\right], \tag{3}$$

where $\mathbf{r}_{1:t}^\pi$ are the firing rates of the decision network units up to time $t$, $v_\phi$ denotes the value function as described below, and the gradient $\nabla_\theta \log \pi_\theta(\mathbf{a}_t|\mathbf{u}_{1:t})$, known as the *eligibility*, [and likewise $\nabla_\theta \Omega_n^\pi(\theta)$] is computed by backpropagation through time (BPTT) (*Rumelhart et al., 1986*) for the selected actions $\mathbf{a}_t$. The sum over rewards in large brackets only runs over $\tau = t, \ldots, T$, which reflects the fact that present actions do not affect past rewards. In this form the terms in the gradient have the intuitive property that they are nonzero only if the actual return deviates from what was predicted by the baseline. It is worth noting that this form of the value function (with access to the selected action) can, in principle, lead to suboptimal policies if the value network's predictions become perfect before the optimal decision policy is learned; we did not find this to be the case in our simulations.

The reward baseline is an important feature in the success of almost all REINFORCE-based algorithms, and is here represented by a second RNN $v_\phi$ with parameters $\phi$ in addition to the decision network $\pi_\theta$ (to be precise, the value function is the readout of the value network). This baseline network, which we call the *value network*, uses the selected actions $\mathbf{a}_{1:t}$ and activity of the decision network $\mathbf{r}_{1:t}^\pi$ to predict the expected return at each time $t = 1, \ldots, T$; the value network also predicts the

expected return at $t = 0$ based on its own initial states, with the understanding that $\mathbf{a}_{1:0} = \emptyset$ and $\mathbf{r}^\pi_{1:0} = \emptyset$ are empty sets. The value network is trained by minimizing a second objective function

$$\mathcal{L}^v(\phi) = \frac{1}{N_{\text{trials}}} \sum_{n=1}^{N_{\text{trials}}} \left[ E_n(\phi) + \Omega^v_n(\phi) \right], \tag{4}$$

$$E_n(\phi) = \frac{1}{T+1} \sum_{t=0}^{T} \left[ \sum_{\tau=t}^{T} \rho_{\tau+1} - v_\phi\left(\mathbf{a}_{1:t}, \mathbf{r}^\pi_{1:t}\right) \right]^2 \tag{5}$$

every $N_{\text{trials}}$ trials, where $\Omega^v_n(\phi)$ denotes any regularization terms for the value network. The necessary gradient $\nabla_\phi E_n(\phi)$ [and likewise $\nabla_\phi \Omega^v_n(\phi)$] is again computed by BPTT.

## Decision and value recurrent neural networks

The policy probability distribution over actions $\pi_\theta(\mathbf{a}_t | \mathbf{u}_{1:t})$ and scalar baseline $v_\phi\left(\mathbf{a}_{1:t}, \mathbf{r}^\pi_{1:t}\right)$ are each represented by an RNN of $N$ firing-rate units $\mathbf{r}^\pi$ and $\mathbf{r}^v$, respectively, where we interpret each unit as the mean firing rate of a group of neurons. In the case where the agent chooses a single action at each time $t$, the activity of the decision network determines $\pi_\theta(\mathbf{a}_t | \mathbf{u}_{1:t})$ through a linear readout followed by softmax normalization:

$$\mathbf{z}_t = W^\pi_{\text{out}} \mathbf{r}^\pi_t + \mathbf{b}^\pi_{\text{out}}, \tag{6}$$

$$\pi_\theta(a_t = k | \mathbf{u}_{1:t}) = \frac{e^{(\mathbf{z}_t)_k}}{\sum_{\ell=1}^{N_a} e^{(\mathbf{z}_t)_\ell}} \tag{7}$$

for $k = 1, \ldots, N_a$. Here $W^\pi_{\text{out}}$ is an $N_a \times N$ matrix of connection weights from the units of the decision network to the $N_a$ linear readouts $\mathbf{z}_t$, and $\mathbf{b}^\pi_{\text{out}}$ are $N_a$ biases. Action selection is implemented by randomly sampling from the probability distribution in *Equation 7*, and constitutes an important difference from previous approaches to training RNNs for cognitive tasks (*Mante et al., 2013*; *Carnevale et al., 2015*; *Song et al., 2016*; *Miconi, 2016*), namely, here the final output of the network (during training) is a specific action, not a graded decision variable. We consider this sampling as an abstract representation of the downstream action selection mechanisms present in the brain, including the role of noise in implicitly realizing stochastic choices with deterministic outputs (*Wang, 2002*, *2008*). Meanwhile, the activity of the value network predicts future returns through a linear readout

$$v_\phi\left(\mathbf{a}_{1:t}, \mathbf{r}^\pi_{1:t}\right) = W^v_{\text{out}} \mathbf{r}^v_t + b^v_{\text{out}}, \tag{8}$$

where $W^v_{\text{out}}$ is an $1 \times N$ matrix of connection weights from the units of the value network to the single linear readout $v_\phi$, and $b^v_{\text{out}}$ is a bias term.

In order to take advantage of recent developments in training RNNs [in particular, addressing the problem of vanishing gradients (*Bengio et al., 1994*)] while retaining intepretability, we use a modified form of Gated Recurrent Units (GRUs) (*Cho et al., 2014*; *Chung et al., 2014*) with a threshold-linear 'f-I' curve $[x]_+ = max(0, x)$ to obtain positive, non-saturating firing rates. Since firing rates in cortex rarely operate in the saturating regime, previous work (*Sussillo et al., 2015*) used an additional regularization term to prevent saturation in common nonlinearities such as the hyperbolic tangent; the threshold-linear activation function obviates such a need. These units are thus leaky, threshold-linear units with dynamic time constants and gated recurrent inputs. The equations that describe their dynamics can be derived by a naïve discretization of the following continuous-time equations for the $N$ currents $\mathbf{x}$ and corresponding rectified-linear firing rates $\mathbf{r}$:

$$\boldsymbol{\lambda} = \text{sigmoid}\left(W^\lambda_{\text{rec}} \mathbf{r} + W^\lambda_{\text{in}} \mathbf{u} + \mathbf{b}^\lambda\right), \tag{9}$$

$$\boldsymbol{\gamma} = \text{sigmoid}\left(W^\gamma_{\text{rec}} \mathbf{r} + W^\gamma_{\text{in}} \mathbf{u} + \mathbf{b}^\gamma\right), \tag{10}$$

$$\frac{\boldsymbol{\tau}}{\boldsymbol{\lambda}} \odot \dot{\mathbf{x}} = -\mathbf{x} + W_{\text{rec}}(\boldsymbol{\gamma} \odot \mathbf{r}) + W_{\text{in}} \mathbf{u} + \mathbf{b} + \sqrt{2\tau\sigma^2_{\text{rec}}}\boldsymbol{\xi}, \tag{11}$$

$$\mathbf{r} = [\mathbf{x}]_+. \tag{12}$$

Here $\dot{\mathbf{x}} = d\mathbf{x}/dt$ is the derivative of $\mathbf{x}$ with respect to time, $\odot$ denotes elementwise multiplication, $\text{sigmoid}(x) = [1 + e^{-x}]^{-1}$ is the logistic sigmoid, $\mathbf{b}^\lambda$, $\mathbf{b}^\gamma$, and $\mathbf{b}$ are biases, $\xi$ are $N$ independent

Gaussian white noise processes with zero mean and unit variance, and $\sigma_{\text{rec}}^2$ controls the size of this noise. The multiplicative gates $\lambda$ dynamically modulate the overall time constant $\tau$ for network units, while the $\gamma$ control the recurrent inputs. The $N \times N$ matrices $W_{\text{rec}}$, $W_{\text{rec}}^\lambda$, and $W_{\text{rec}}^\gamma$ are the recurrent weight matrices, while the $N \times N_{\text{in}}$ matrices $W_{\text{in}}$, $W_{\text{in}}^\lambda$, and $W_{\text{in}}^\gamma$ are connection weights from the $N_{\text{in}}$ inputs $\mathbf{u}$ to the $N$ units of the network. We note that in the case where $\lambda \to 1$ and $\gamma \to 1$ the equations reduce to 'simple' leaky threshold-linear units without the modulation of the time constants or gating of inputs. We constrain the recurrent connection weights (*Song et al., 2016*) so that the overall connection probability is $p_c$; specifically, the number of incoming connections for each unit, or in-degree $K$, was set to $K = p_c N$ (see *Table 1* for a list of all parameters).

The result of discretizing *Equations 9–12*, as well as details on initializing the network parameters, are given in Materials and methods. We successfully trained networks with time steps $\Delta t = 1\text{ms}$, but for computational convenience all of the networks in this work were trained and run with $\Delta t = 10\text{ms}$. We note that, for typical tasks in systems neuroscience lasting on the order of several seconds, this already implies trials lasting hundreds of time steps. Unless noted otherwise in the text, all networks were trained using the parameters listed in *Table 1*.

While the inputs to the decision network $\pi_\theta$ are determined by the environment, the value network always receives as inputs the activity of the decision network $\mathbf{r}^\pi$, together with information about which actions were actually selected at each time step (*Figure 1B*). The value network serves two purposes: first, the output of the value network is used as the baseline in the REINFORCE gradient, *Equation 3*, to reduce the variance of the gradient estimate (*Williams, 1992*; *Baird and Moore, 1999*; *Baxter and Bartlett, 2001*; *Peters and Schaal, 2008*); second, since policy gradient reinforcement learning does not explicitly use a value function but value information is nevertheless implicitly contained in the policy, the value network serves as an explicit and potentially nonlinear readout of this information. In situations where expected reward is closely related to confidence, this may explain, for example, certain disassociations between perceptual decisions and reports of the associated confidence (*Lak et al., 2014*).

A reward baseline, which allows the decision network to update its parameters based on a relative quantity akin to prediction error (*Schultz et al., 1997*; *Bayer and Glimcher, 2005*) rather than absolute reward magnitude, is essential to many learning schemes, especially those based on REINFORCE. Indeed, it has been suggested that in general such a baseline should be not only task-specific but stimulus (task-condition)-specific (*Frémaux et al., 2010*; *Engel et al., 2015*; *Miconi, 2016*), and that this information may be represented in OFC (*Wallis, 2007*) or basal ganglia (*Doya, 2000*). Previous schemes, however, did not propose how this baseline *critic* may be instantiated, instead implementing it algorithmically. Here we use a simple neural implementation of the baseline that automatically depends on the stimulus and thus does not require the learning system to have access to the true trial type, which in general is not known with certainty to the agent.

**Table 1.** Parameters for reward-based recurrent neural network training. Unless noted otherwise in the text, networks were trained and run with the parameters listed here.

| Parameter | Symbol | Default value |
|---|---|---|
| Learning rate | $\eta$ | 0.004 |
| Maximum gradient norm | $\Gamma$ | 1 |
| Size of decision/value network | $N$ | 100 |
| Connection probability (decision network) | $p_c^\pi$ | 0.1 |
| Connection probability (value network) | $p_c^v$ | 1 |
| Time step | $\Delta t$ | 10 ms |
| Unit time constant | $\tau$ | 100 ms |
| Recurrent noise | $\sigma_{\text{rec}}^2$ | 0.01 |
| Initial spectral radius for recurrent weights | $\rho_0$ | 2 |
| Number of trials per gradient update | $N_{\text{trials}}$ | # of task conditions |

## Tasks with simple input-output mappings

The training procedure described in the previous section can be used for a variety of tasks, and results in networks that qualitatively reproduce both behavioral and electrophysiological findings from experiments with behaving animals. For the example perceptual decision-making task above, the trained network learns to integrate the sensory evidence to make the correct decision about which of two noisy inputs is larger (*Figure 1C*). This and additional networks trained for the same task were able to reach the target performance in ~7000 trials starting from completely random connection weights, and moreover the networks learned the 'core' task after ~2000 trials (*Figure 1—figure supplement 1*). As with monkeys performing the task, longer stimulus durations allow the network to improve its performance by continuing to integrate the incoming sensory evidence (*Wang, 2002*; *Kiani et al., 2008*). Indeed, the output of the value network shows that the expected reward (in this case equivalent to confidence) is modulated by stimulus difficulty (*Figure 1E*). Prior to the onset of the stimulus, the expected reward is the same for all trial conditions and approximates the overall reward rate; incoming sensory evidence then allows the network to distinguish its chances of success.

Sorting the activity of individual units in the network by the signed coherence (the strength of the evidence, with negative values indicating evidence for L and positive for R) also reveals coherence-dependent ramping activity (*Figure 1D*) as observed in neural recordings from numerous perceptual decision-making experiments, e.g., *Roitman and Shadlen (2002)*. This pattern of activity illustrates why a nonlinear readout by the value network is useful: expected return is computed by performing an 'absolute value'-like operation on the accumulated evidence (plus shifts), as illustrated by the overlap of the expected return for positive and negative-coherence trials (*Figure 1E*).

The reaction time as a function of coherence in the reaction-time version of the same task, in which the go cue coincides with the time of stimulus onset, is also shown in *Figure 1—figure supplement 2* and may be compared, e.g., to *Wang (2002)*; *Mazurek et al. (2003)*; *Wong and Wang (2006)*. We note that in many neural models [e.g., *Wang (2002)*; *Wong and Wang (2006)*] a 'decision' is made when the output reaches a fixed threshold. Indeed, when networks are trained using supervised learning (*Song et al., 2016*), the decision threshold is imposed retroactively and has no meaning during training; since the outputs are continuous, the speed-accuracy tradeoff is also learned in the space of continuous error signals. Here, the time at which the network commits to a decision is unambiguously given by the time at which the selected action is L or R. Thus the appropriate speed-accuracy tradeoff is learned in the space of concrete actions, illustrating the desirability of using reward-based training of RNNs when modeling reaction-time tasks. Learning curves for this and additional networks trained for the same reaction-time task are shown in *Figure 1—figure supplement 3*.

In addition to the example task from the previous section, we trained networks for three well-known behavioral paradigms in which the correct, or optimal, behavior is (pre-)determined on each trial by the task condition alone. Similar tasks have previously been addressed with several different forms of supervised learning, including FORCE (*Sussillo and Abbott, 2009*; *Carnevale et al., 2015*), Hessian-free (*Martens and Sutskever, 2011*; *Mante et al., 2013*; *Barak et al., 2013*), and stochastic gradient descent (*Pascanu et al., 2013b*; *Song et al., 2016*), so that the results shown in *Figure 2* are presented as confirmation that the same tasks can also be learned using reward feedback on definite actions alone. For all three tasks the pre-stimulus fixation period was 750 ms; the networks had to maintain fixation until the start of a 500 ms 'decision' period, which was indicated by the extinction of the fixation cue. At this time the network was required to choose one of two alternatives to indicate its decision and receive a reward of +1 for a correct response and 0 for an incorrect response; otherwise, the networks received a reward of −1.

The context-dependent integration task (*Figure 2A*) is based on *Mante et al. (2013)*, in which monkeys were required to integrate one type of stimulus (the motion or color of the presented dots) while ignoring the other depending on a context cue. In training the network, we included both the 750 ms stimulus period and 300–1500 ms delay period following stimulus presentation. The delay consisted of 300 ms followed by a variable duration drawn from an exponential distribution with mean 300 ms and truncated at a maximum of 1200 ms. The network successfully learned to perform the task, which is reflected in the psychometric functions showing the percentage of trials on which the network chose R as a function of the signed motion and color coherences, where motion and

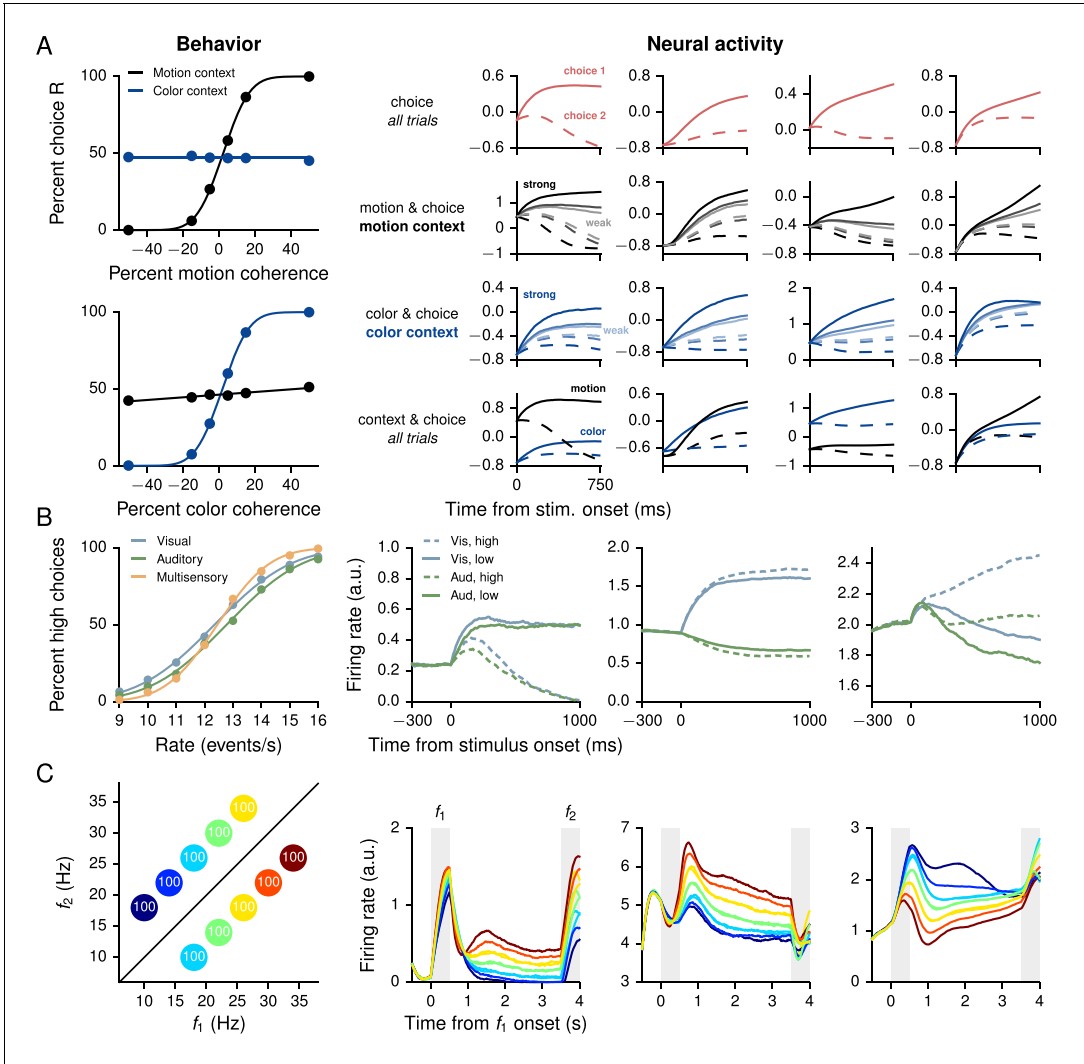

**Figure 2.** Performance and neural activity of RNNs trained for 'simple' cognitive tasks in which the correct response depends only on the task condition. Left column shows behavioral performance, right column shows mixed selectivity for task parameters of example units in the decision network. (**A**) Context-dependent integration task (*Mante et al., 2013*). Left: Psychometric curves show the percentage of R choices as a function of the signed 'motion' and 'color' coherences in the motion (black) and color (blue) contexts. Right: Normalized firing rates of examples units sorted by different combinations of task parameters exhibit mixed selectivity. Firing rates were normalized by mean and standard deviation computed over the responses of all units, times, and trials. Solid and dashed lines indicate choice 1 (same as preferred direction of unit) and choice 2 (non-preferred), respectively. For motion and choice and color and choice, dark to light corresponds to high to low motion and color coherence, respectively. (**B**) Multisensory integration task (*Raposo et al., 2012*, *2014*). Left: Psychometric curves show the percentage of high choices as a function of the event rate, for visual only (blue), auditory only (green), and multisensory (orange) trials. Improved performance on multisensory trials shows that the network learns to combine the two sources of information in accordance with *Equation 13*. Right: Sorted activity on visual only and auditory only trials for units selective for choice (high vs. low, left), modality [visual (vis) vs. auditory (aud), middle], and both (right). Error trials were excluded. (**C**) Parametric working memory task (*Romo et al., 1999*). Left: Percentage of correct responses for different combinations of $f_1$ and $f_2$. The conditions are colored here and in the right panels according to the first stimulus (base frequency) $f_1$; due to the overlap in the values of $f_1$, the 10 task conditions are represented by seven distinct colors. Right: Activity of example decision network units sorted by $f_1$. The first two units are positively tuned to $f_1$ during the delay period, while the third unit is negatively tuned.

The following figure supplements are available for figure 2:

**Figure supplement 1.** Learning curves for the context-dependent integration task.

**Figure supplement 2.** Learning curves for the multisensory integration task.

**Figure supplement 3.** Learning curves for the parametric working memory task.

color indicate the two sources of noisy information and the sign is positive for R and negative for L (*Figure 2A*, left). As in electrophysiological recordings, units in the decision network show mixed selectivity when sorted by different combinations of task variables (*Figure 2A*, right). Learning curves for this and additional networks trained for the task are shown in *Figure 2—figure supplement 1*.

The multisensory integration task (*Figure 2B*) is based on *Raposo et al. (2012, 2014)*, in which rats used visual flashes and auditory clicks to determine whether the event rate was higher or lower than a learned threshold of 12.5 events per second. When both modalities were presented, they were congruent, which implied that the rats could improve their performance by combining information from both sources. As in the experiment, the network was trained with a 1000 ms stimulus period, with inputs whose magnitudes were proportional (both positively and negatively) to the event rate. For this task the input connection weights $W_{\text{in}}$, $W_{\text{in}}^{\text{A}}$, and $W_{\text{in}}^{\gamma}$ were initialized so that a third of the $N = 150$ decision network units received visual inputs only, another third auditory inputs only, and the remaining third received neither. As shown in the psychometric function (percentage of high choices as a function of event rate, *Figure 2B*, left), the trained network exhibits multisensory enhancement in which performance on multisensory trials was better than on single-modality trials. Indeed, as for rats, the results are consistent with optimal combination of the two modalities,

$$\frac{1}{\sigma_{\text{visual}}^2} + \frac{1}{\sigma_{\text{auditory}}^2} \approx \frac{1}{\sigma_{\text{multisensory}}^2}, \tag{13}$$

where $\sigma_{\text{visual}}^2$, $\sigma_{\text{auditory}}^2$, and $\sigma_{\text{multisensory}}^2$ are the variances obtained from fits of the psychometric functions to cumulative Gaussian functions for visual only, auditory only, and multisensory (both visual and auditory) trials, respectively (*Table 2*). As observed in electrophysiological recordings, moreover, decision network units exhibit a range of tuning to task parameters, with some selective to choice and others to modality, while many units showed mixed selectivity to all task variables (*Figure 2B*, right). Learning curves for this and additional networks trained for the task are shown in *Figure 2—figure supplement 2*.

The parametric working memory task (*Figure 2C*) is based on the vibrotactile frequency discrimination task of *Romo et al. (1999)*, in which monkeys were required to compare the frequencies of two temporally separated stimuli to determine which was higher. For network training, the task epochs consisted of a 500 ms base stimulus with 'frequency' $f_1$, a 2700–3300 ms delay, and a 500 ms comparison stimulus with frequency $f_2$; for the trials shown in *Figure 2C* the delay was always 3000 ms as in the experiment. During the decision period, the network had to indicate which stimulus was higher by choosing $f_1 < f_2$ or $f_1 > f_2$. The stimuli were constant inputs with amplitudes proportional (both positively and negatively) to the frequency. For this task we set the learning rate to $\eta = 0.002$; the network successfully learned to perform the task (*Figure 2C*, left), and the individual units of the network, when sorted by the first stimulus (base frequency) $f_1$, exhibit highly heterogeneous activity (*Figure 2C*, right) characteristic of neurons recorded in the prefrontal cortex of monkeys performing

**Table 2.** Psychophysical thresholds $\sigma_{\text{visual}}$, $\sigma_{\text{auditory}}$, and $\sigma_{\text{multisensory}}$ obtained from fits of cumulative Gaussian functions to the psychometric curves in visual only, auditory only, and multisensory trials in the multisensory integration task, for six networks trained from different random initializations (first row, bold: network from main text, cf. *Figure 2B*). The last two columns show evidence of 'optimal' multisensory integration according to *Equation 13* (*Raposo et al., 2012*).

| $\sigma_{\text{visual}}$ | $\sigma_{\text{auditory}}$ | $\sigma_{\text{multisensory}}$ | $\dfrac{1}{\sigma_{\text{visual}}^2} + \dfrac{1}{\sigma_{\text{auditory}}^2}$ | $\dfrac{1}{\sigma_{\text{multisensory}}^2}$ |
|---|---|---|---|---|
| 2.124 | 2.099 | 1.451 | 0.449 | 0.475 |
| 2.107 | 2.086 | 1.448 | 0.455 | 0.477 |
| 2.276 | 2.128 | 1.552 | 0.414 | 0.415 |
| 2.118 | 2.155 | 1.508 | 0.438 | 0.440 |
| 2.077 | 2.171 | 1.582 | 0.444 | 0.400 |
| 2.088 | 2.149 | 1.480 | 0.446 | 0.457 |

the task (*Machens et al., 2010*). Learning curves for this and additional networks trained for the task are shown in *Figure 2—figure supplement 3*.

Additional comparisons can be made between the model networks shown in *Figure 2* and the neural activity observed in behaving animals, for example state-space analyses as in *Mante et al. (2013)*, *Carnevale et al. (2015)*, or *Song et al. (2016)*. Such comparisons reveal that, as found previously in studies such as *Barak et al. 2013)*, the model networks exhibit many, but not all, features present in electrophysiological recordings. *Figure 2* and the following make clear, however, that RNNs trained with reward feedback alone can already reproduce the mixed selectivity characteristic of neural populations in higher cortical areas (*Rigotti et al., 2010*, *2013*), thereby providing a valuable platform for future investigations of how such complex representations are learned.

## Confidence and perceptual decision-making

All of the tasks in the previous section have the property that the correct response on any single trial is a function only of the task condition, and, in particular, does not depend on the network's state during the trial. In a postdecision wager task (*Kiani and Shadlen, 2009*), however, the optimal decision depends on the animal's (agent's) estimate of the probability that its decision is correct, i.e., its confidence. As can be seen from the results, on a trial-by-trial basis this is not the same as simply determining the stimulus difficulty (a combination of stimulus duration and coherence); this makes it difficult to train with standard supervised learning, which requires a pre-determined target output for the network to reproduce; instead, we trained an RNN to perform the task by maximizing overall reward. This task extends the simple perceptual decision-making task (*Figure 1A*) by introducing a 'sure' option that is presented during a 1200–1800 ms delay period on a random half of the trials; selecting this option results in a reward that is 0.7 times the size of the reward obtained when correctly choosing L or R. As in the monkey experiment, the network receives no information indicating whether or not a given trial will contain a sure option until the middle of the delay period after stimulus offset, thus ensuring that the network makes a decision about the stimulus on all trials (*Figure 3A*). For this task the input connection weights $W_{\mathrm{in}}$, $W_{\mathrm{in}}^{\lambda}$, and $W_{\mathrm{in}}^{\gamma}$ were initialized so that half the units received information about the sure target while the other half received evidence for L and R. All units initially received fixation input.

The key behavioral features found in *Kiani and Shadlen (2009)*; *Wei and Wang (2015)* are reproduced in the trained network, namely the network opted for the sure option more frequently when the coherence was low or stimulus duration short (*Figure 3B*, left); and when the network was presented with a sure option but waived it in favor of choosing L or R, the performance was better than on trials when the sure option was not presented (*Figure 3B*, right). The latter observation is taken as indication that neither monkeys nor trained networks choose the sure target on the basis of stimulus difficulty alone but based on their internal sense of uncertainty on each trial.

*Figure 3C* shows the activity of an example network unit, sorted by whether the decision was the unit's preferred or nonpreferred target (as determined by firing rates during the stimulus period on all trials), for both non-wager and wager trials. In particular, on trials in which the sure option was chosen, the firing rate is intermediate compared to trials on which the network made a decision by choosing L or R. Learning curves for this and additional networks trained for the task are shown in *Figure 3—figure supplement 1*.

## Value-based economic choice task

We also trained networks to perform the simple economic choice task of *Padoa-Schioppa and Assad (2006)* and examined the activity of the *value*, rather than decision, network. The choice patterns of the networks were modulated only by varying the reward contingencies (*Figure 4A*, upper and lower). We note that, on each trial there is a 'correct' answer in the sense that there is a choice which results in greater reward. In contrast to the previous tasks, however, information regarding whether an answer is correct in this sense is not contained in the inputs but rather in the association between inputs and rewards. This distinguishes the task from the cognitive tasks discussed in previous sections: although the task can be transformed into a cognitive-type task by providing the associated rewards as inputs, training in this manner conflates external with 'internal,' learned inputs.

Each trial began with a 750 ms fixation period; the offer, which indicated the 'juice' type and amount for the left and right choices, was presented for 1000–2000 ms, followed by a 750 ms

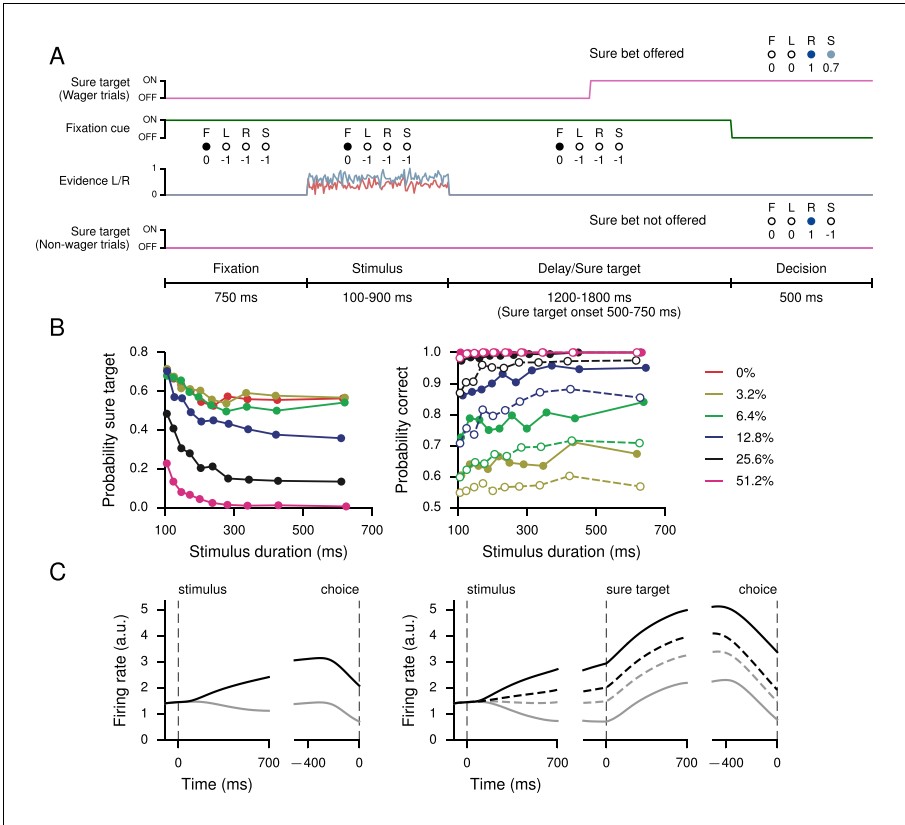

**Figure 3.** Perceptual decision-making task with postdecision wagering, based on *Kiani and Shadlen (2009)*. (**A**) Task structure. On a random half of the trials, a sure option is presented during the delay period, and on these trials the network has the option of receiving a smaller (compared to correctly choosing L or R) but certain reward by choosing the sure option (S). The stimulus duration, delay, and sure target onset time are the same as in *Kiani and Shadlen 2009*. (**B**) Probability of choosing the sure option (left) and probability correct (right) as a function of stimulus duration, for different coherences. Performance is higher for trials on which the sure option was offered but waived in favor of L or R (filled circles, solid), compared to trials on which the sure option was not offered (open circles, dashed). (**C**) Activity of an example decision network unit for non-wager (left) and wager (right) trials, sorted by whether the presented evidence was toward the unit's preferred (black) or nonpreferred (gray) target as determined by activity during the stimulus period on all trials. Dashed lines show activity for trials in which the sure option was chosen.

The following figure supplement is available for figure 3:

**Figure supplement 1.** Learning curves for the postdecision wager task.

decision period during which the network was required to indicate its decision. In the upper panel of *Figure 4A* the indifference point was set to 1A = 2.2B during training, which resulted in 1A = 2.0B when fit to a cumulative Gaussian (*Figure 4—figure supplement 1*), while in the lower panel it was set to 1A = 4.1B during training and resulted in 1A = 4.0B (*Figure 4—figure supplement 2*). The basic unit of reward, i.e., 1B, was 0.1. For this task we increased the initial value of the value network's input weights, $W_{\mathrm{in}}^v$, by a factor of 10 to drive the value network more strongly.

Strikingly, the activity of units in the value network $v_\phi$ exhibits similar types of tuning to task variables as observed in the orbitofrontal cortex of monkeys, with some units (roughly 20% of active units) selective to chosen value, others (roughly 60%, for both A and B) to offer value, and still others (roughly 20%) to choice alone as defined in *Padoa-Schioppa and Assad (2006)* (*Figure 4B*). The decision network also contained units with a diversity of tuning. Learning curves for this and additional networks trained for the task are shown in *Figure 4—figure supplement 3*. We emphasize that no changes were made to the network architecture for this value-based economic choice task.

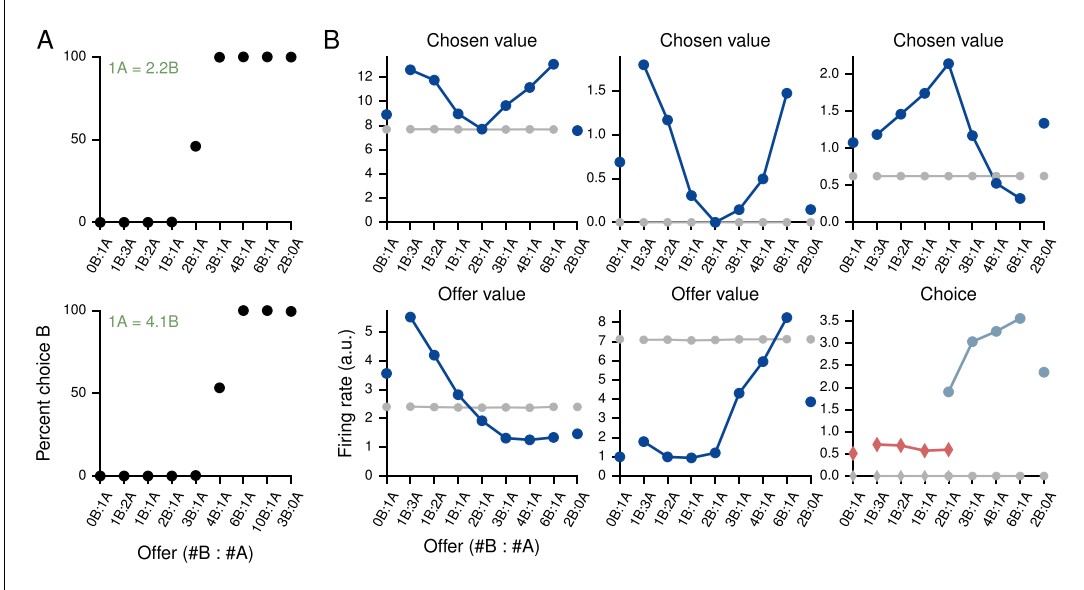

**Figure 4.** Value-based economic choice task (*Padoa-Schioppa and Assad, 2006*). (**A**) Choice pattern when the reward contingencies are indifferent for roughly 1 'juice' of A and 2 'juices' of B (upper) or 1 juice of A and 4 juices of B (lower). (**B**) Mean activity of example value network units during the pre-choice period, defined here as the period 500 ms before the decision, for the 1A = 2B case. Units in the value network exhibit diverse selectivity as observed in the monkey orbitofrontal cortex. For 'choice' (last panel), trials were separated into choice **A** (red diamonds) and choice **B** (blue circles).

The following figure supplements are available for figure 4:

**Figure supplement 1.** Fit of cumulative Gaussian with parameters $\mu$, $\sigma$ to the choice pattern in *Figure 4* (upper), and the deduced indifference point $n_B^*/n_A^* = (1 + \mu)/(1 - \mu)$.

**Figure supplement 2.** Fit of cumulative Gaussian with parameters $\mu$, $\sigma$ to the choice pattern in *Figure 4A* (lower), and the deduced indifference point $n_B^*/n_A^* = (1 + \mu)/(1 - \mu)$.

**Figure supplement 3.** Learning curves for the value-based economic choice task.

Instead, the same scheme shown in *Figure 1B*, in which the value network is responsible for predicting future rewards to guide learning but is *not* involved in the execution of the policy, gave rise to the pattern of neural activity shown in *Figure 4B*.

## Discussion

In this work we have demonstrated reward-based training of recurrent neural networks for both cognitive and value-based tasks. Our main contributions are twofold: first, our work expands the range of tasks and corresponding neural mechanisms that can be studied by analyzing model recurrent neural networks, providing a unified setting in which to study diverse computations and compare to electrophysiological recordings from behaving animals; second, by explicitly incorporating reward into network training, our work makes it possible in the future to more directly address the question of reward-related processes in the brain, for instance the role of value representation that is essential for learning, but not executing, a task.

To our knowledge, the specific form of the baseline network inputs used in this work has not been used previously in the context of recurrent policy gradients; it combines ideas from *Wierstra et al. (2009)* where the baseline network received the same inputs as the decision network in addition to the selected actions, and *Ranzato et al. (2016)*, where the baseline was implemented as a simple linear regressor of the activity of the decision network, so that the decision and value networks effectively shared the same recurrent units. Indeed, the latter architecture is quite common in machine learning applications (*Mnih et al., 2016*), and likewise, for some of the simpler tasks

considered here, models with a baseline consisting of a linear readout of the selected actions and decision network activity could be trained in comparable (but slightly longer) time (*Figure 1—figure supplement 4*). The question of whether the decision and value networks ought to share the same recurrent network parallels ongoing debate over whether choice and confidence are computed together or if certain areas such as OFC compute confidence signals locally, though it is clear that such 'meta-cognitive' representations can be found widely in the brain (*Lak et al., 2014*). Computationally, the distinction is expected to be important when there are nonlinear computations required to determine expected return that are not needed to implement the policy, as illustrated in the perceptual decision-making task (*Figure 1*).

Interestingly, a separate value network to represent the baseline suggests an explicit role for value representation in the brain that is essential for learning a task (equivalently, when the environment is changing), but not for executing an already learned task, as is sometimes found in experiments (*Turner and Desmurget, 2010*; *Schoenbaum et al., 2011*; *Stalnaker et al., 2015*). Since an accurate baseline dramatically improves learning but is not *required*—the algorithm is less reliable and takes many samples to converge with a constant baseline, for instance—this baseline network hypothesis for the role of value representation may account for some of the subtle yet broad learning deficits observed in OFC-lesioned animals (*Wallis, 2007*). Moreover, since expected reward is closely related to decision confidence in many of the tasks considered, a value network that nonlinearly reads out confidence information from the decision network is consistent with experimental findings in which OFC inactivation affects the ability to report confidence but not decision accuracy (*Lak et al., 2014*).

Our results thus support the actor-critic picture for reward-based learning, in which one circuit directly computes the policy to be followed, while a second structure, receiving projections from the decision network as well as information about the selected actions, computes expected future reward to guide learning. Actor-critic models have a rich history in neuroscience, particularly in studies of the basal ganglia (*Houk et al., 1995*; *Dayan and Balleine, 2002*; *Joel et al., 2002*; *O'Doherty et al., 2004*; *Takahashi et al., 2008*; *Maia, 2010*), and it is interesting to note that there is some experimental evidence that signals in the striatum are more suitable for direct policy search rather than for updating action values as an intermediate step, as would be the case for purely value function-based approaches to computing the decision policy (*Li and Daw, 2011*; *Niv and Langdon, 2016*). Moreover, although we have used a single RNN each to represent the decision and value modules, using 'deep,' multilayer RNNs may increase the representational power of each module (*Pascanu et al., 2013a*). For instance, more complex tasks than considered in this work may require hierarchical feature representation in the decision network, and likewise value networks can use a combination of the different features [including raw sensory inputs (*Wierstra et al., 2009*)] to predict future reward. Anatomically, the decision networks may correspond to circuits in dorsolateral prefrontal cortex, while the value networks may correspond to circuits in OFC (*Schultz et al., 2000*; *Takahashi et al., 2011*) or basal ganglia (*Hikosaka et al., 2014*). This architecture also provides a useful example of the hypothesis that various areas of the brain effectively optimize different cost functions (*Marblestone et al., 2016*): in this case, the decision network maximizes reward, while the value network minimizes the prediction error for future reward.

As in many other supervised learning approaches used previously to train RNNs (*Mante et al., 2013*; *Song et al., 2016*), the use of BPTT to compute the gradients (in particular, the eligibility) make our 'plasticity rule' not biologically plausible. As noted previously (*Zipser and Andersen, 1988*), it is indeed somewhat surprising that the activity of the resulting networks nevertheless exhibit many features found in neural activity recorded from behaving animals. Thus our focus has been on learning from realistic feedback signals provided by the environment but not on its physiological implementation. Still, recent work suggests that exact backpropagation is not necessary and can even be implemented in 'spiking' stochastic units (*Lillicrap et al., 2016*), and that approximate forms of backpropagation and SGD can be implemented in a biologically plausible manner (*Scellier and Bengio, 2016*), including both spatially and temporally asynchronous updates in RNNs (*Jaderberg et al., 2016*). Such ideas require further investigation and may lead to effective yet more neurally plausible methods for training model neural networks.

Recently, *Miconi (2016)* used a 'node perturbation'-based (*Fiete and Seung, 2006*; *Fiete et al., 2007*; *Hoerzer et al., 2014*) algorithm with an error signal at the end of each trial to train RNNs for several cognitive tasks, and indeed, node perturbation is closely related to the REINFORCE

algorithm used in this work. On one hand, the method described in *Miconi (2016)* is more biologically plausible in the sense of not requiring gradients computed via backpropagation through time as in our approach; on the other hand, in contrast to the networks in this work, those in *Miconi (2016)* did not 'commit' to a discrete action and thus the error signal was a graded quantity. In this and other works (*Frémaux et al., 2010*), moreover, the prediction error was computed by algorithmically keeping track of a stimulus (task condition)-specific running average of rewards. Here we used a concrete scheme (namely a value network) for approximating the average that automatically depends on the stimulus, without requiring an external learning system to maintain a separate record for each (true) trial type, which is not known by the agent with certainty.

One of the advantages of the REINFORCE algorithm for policy gradient reinforcement learning is that direct supervised learning can also be mixed with reward-based learning, by including only the eligibility term in *Equation 3* without modulating by reward (*Mnih et al., 2014*), i.e., by maximizing the log-likelihood of the desired actions. Although all of the networks in this work were trained from reward feedback only, it will be interesting to investigate this feature of the REINFORCE algorithm. Another advantage, which we have not exploited here, is the possibility of learning policies for continuous action spaces (*Peters and Schaal, 2008*; *Wierstra et al., 2009*); this would allow us, for example, to model arbitrary saccade targets in the perceptual decision-making task, rather than limiting the network to discrete choices.

We have previously emphasized the importance of incorporating biological constraints in the training of neural networks (*Song et al., 2016*). For instance, neurons in the mammalian cortex have purely excitatory or inhibitory effects on other neurons, which is a consequence of Dale's Principle for neurotransmitters (*Eccles et al., 1954*). In this work we did not include such constraints due to the more complex nature of our rectified GRUs (*Equations 9–12*); in particular, the units we used are capable of dynamically modulating their time constants and gating their recurrent inputs, and we therefore interpreted the firing rate units as a mixture of both excitatory and inhibitory populations. Indeed, these may implement the 'reservoir of time constants' observed experimentally (*Bernacchia et al., 2011*). In the future, however, comparison to both model spiking networks and electrophysiological recordings will be facilitated by including more biological realism, by explicitly separating the roles of excitatory and inhibitory units (*Mastrogiuseppe and Ostojic, 2016*). Moreover, since both the decision and value networks are obtained by minimizing an objective function, additional regularization terms can be easily included to obtain networks whose activity is more similar to neural recordings (*Sussillo et al., 2015*; *Song et al., 2016*).

Finally, one of the most appealing features of RNNs trained to perform many tasks is their ability to provide insights into neural computation in the brain. However, methods for revealing neural mechanisms in such networks remain limited to state-space analysis (*Sussillo and Barak, 2013*), which in particular does not reveal how the synaptic connectivity leads to the dynamics responsible for implementing the higher-level decision policy. General and systematic methods for analyzing trained networks are still needed and are the subject of ongoing investigation. Nevertheless, reward-based training of RNNs makes it more likely that the resulting networks will correspond closely to biological networks observed in experiments with behaving animals. We expect that the continuing development of tools for training model neural networks in neuroscience will thus contribute novel insights into the neural basis of animal cognition.

## Materials and methods

### Policy gradient reinforcement learning with RNNs

Here we review the application of the REINFORCE algorithm for policy gradient reinforcement learning to recurrent neural networks (*Williams, 1992*; *Baird and Moore, 1999*; *Sutton et al., 2000*; *Baxter and Bartlett, 2001*; *Peters and Schaal, 2008*; *Wierstra et al., 2009*). In particular, we provide a careful derivation of *Equation 3* following, in part, the exposition in *Zaremba and Sutskever (2016)*.

Let $H_{\mu:t}$ be the sequence of interactions between the environment and agent (i.e., the environmental states, observables, and agent actions) that results in the environment being in state $s_{t+1}$ at time $t + 1$ starting from state $s_\mu$ at time $\mu$:

$$H_{\mu:t} = (\mathbf{s}_{\mu+1:t+1}, \mathbf{u}_{\mu:t}, \mathbf{a}_{\mu:t}). \tag{14}$$

For notational convenience in the following, we adopt the convention that, for the special case of $\mu = 0$, the history $H_{0:t}$ includes the initial state $\mathbf{s}_0$ and excludes the meaningless inputs $\mathbf{u}_0$, which are not seen by the agent:

$$H_{0:t} = (\mathbf{s}_{0:t+1}, \mathbf{u}_{1:t}, \mathbf{a}_{0:t}). \tag{15}$$

When $t = 0$, it is also understood that $\mathbf{u}_{1:0} = \varnothing$, the empty set. A full history, or a trial, is thus denoted as

$$H \equiv H_{0:T} = (\mathbf{s}_{0:T+1}, \mathbf{u}_{1:T}, \mathbf{a}_{0:T}), \tag{16}$$

where $T$ is the end of the trial. Here we only consider the episodic, 'finite-horizon' case where $T$ is finite, and since different trials can have different durations, we take $T$ to be the maximum length of a trial in the task. The reward $\rho_{t+1}$ at time $t+1$ following actions $\mathbf{a}_t$ (we use $\rho$ to distinguish it from the firing rates $\mathbf{r}$ of the RNNs) is determined by this history, which we sometimes indicate explicitly by writing

$$\rho_{t+1} = \rho_{t+1}(H_{0:t}). \tag{17}$$

As noted in the main text, we adopt the convention that the agent performs actions at $t = 0, \ldots, T$ and receives rewards at $t = 1, \ldots, T+1$ to emphasize that rewards follow the actions and are jointly determined with the next state (**Sutton and Barto, 1998**). For notational simplicity, here and elsewhere we assume that any discount factor is already included in $\rho_{t+1}$, i.e., in all places where the reward appears we consider $\rho_{t+1} \to e^{-t/\tau_{\mathrm{reward}}} \rho_{t+1}$, where $\tau_{\mathrm{reward}}$ is the time constant for discounting future rewards (**Doya, 2000**); we included temporal discounting only for the reaction-time version of the simple perceptual decision-making task (**Figure 1—figure supplement 2**), where we set $\tau_{\mathrm{reward}} = 10\mathrm{s}$. For the remaining tasks, $\tau_{\mathrm{reward}} = \infty$.

Explicitly, a trial $H_{0:T}$ comprises the following. At time $t = 0$, the environment is in state $\mathbf{s}_0$ with probability $\mathcal{E}(\mathbf{s}_0)$. The agent initially chooses a set of actions $\mathbf{a}_0$ with probability $\pi_\theta(\mathbf{a}_0)$, which is determined by the parameters of the decision network, in particular the initial conditions $\mathbf{x}_0$ and readout weights $W_{\mathrm{out}}^\pi$ and biases $\mathbf{b}_{\mathrm{out}}^\pi$ (**Equation 6**). At time $t = 1$, the environment, depending on its previous state $\mathbf{s}_0$ and the agent's actions $\mathbf{a}_0$, transitions to state $\mathbf{s}_1$ with probability $\mathcal{E}(\mathbf{s}_1|\mathbf{s}_0, \mathbf{a}_0)$. The history up to this point is $H_{0:0} = (\mathbf{s}_{0:1}, \varnothing, \mathbf{a}_{0:0})$, where $\varnothing$ indicates that no inputs have yet been seen by the network. The environment also generates reward $\rho_1$, which depends on this history, $\rho_1 = \rho_1(H_{0:0})$. From state $\mathbf{s}_1$ the environment generates observables (inputs to the agent) $\mathbf{u}_1$ with a distribution given by $\mathcal{E}(\mathbf{u}_1|\mathbf{s}_1)$. In response, the agent, depending on the inputs $\mathbf{u}_1$ it receives from the environment, chooses the set of actions $\mathbf{a}_1$ according to the distribution $\pi_\theta(\mathbf{a}_1|\mathbf{u}_{1:1}) = \pi_\theta(\mathbf{a}_1|\mathbf{u}_1)$. The environment, depending on its previous states $\mathbf{s}_{0:1}$ and the agent's previous actions $\mathbf{a}_{0:1}$, then transitions to state $\mathbf{s}_2$ with probability $\mathcal{E}(\mathbf{s}_2|\mathbf{s}_{0:1}, \mathbf{a}_{0:1})$. Thus $H_{0:1} = (\mathbf{s}_{0:2}, \mathbf{u}_{1:1}, \mathbf{a}_{0:1})$. Iterating these steps, the history at time $t$ is therefore given by **Equation 15**, while a full history is given by **Equation 16**.

The probability $p_\theta(H_{0:\tau})$ of a particular sub-history $H_{0:\tau}$ up to time $\tau$ occurring, under the policy $\pi_\theta$ parametrized by $\theta$, is given by

$$p_\theta(H_{0:\tau}) = \left[\prod_{t=1}^{\tau} \mathcal{E}(\mathbf{s}_{t+1}|\mathbf{s}_{0:t}, \mathbf{a}_{0:t})\pi_\theta(\mathbf{a}_t|\mathbf{u}_{1:t})\mathcal{E}(\mathbf{u}_t|\mathbf{s}_t)\right]\mathcal{E}(\mathbf{s}_1|\mathbf{s}_0, \mathbf{a}_0)\pi_\theta(\mathbf{a}_0)\mathcal{E}(\mathbf{s}_0). \tag{18}$$

In particular, the probability $p_\theta(H)$ of a history $H = H_{0:T}$ occurring is

$$p_\theta(H) = \left[\prod_{t=1}^{T} \mathcal{E}(\mathbf{s}_{t+1}|\mathbf{s}_{0:t}, \mathbf{a}_{0:t})\pi_\theta(\mathbf{a}_t|\mathbf{u}_{1:t})\mathcal{E}(\mathbf{u}_t|\mathbf{s}_t)\right]\mathcal{E}(\mathbf{s}_1|\mathbf{s}_0, \mathbf{a}_0)\pi_\theta(\mathbf{a}_0)\mathcal{E}(\mathbf{s}_0). \tag{19}$$

A key ingredient of the REINFORCE algorithm is that the policy parameters only indirectly affect the environment through the agent's actions. The logarithmic derivatives of **Equation 18** with respect to the parameters $\theta$ therefore do not depend on the unknown (to the agent) environmental dynamics contained in $\mathcal{E}$, i.e.,

$$\nabla_\theta \log p_\theta(H_{0:\tau}) = \sum_{t=0}^{\tau} \nabla_\theta \log \pi_\theta(\mathbf{a}_t | \mathbf{u}_{1:t}), \tag{20}$$

with the understanding that $\mathbf{u}_{1:0} = \varnothing$ (the empty set) and therefore $\pi_\theta(\mathbf{a}_0 | \mathbf{u}_{1:0}) = \pi_\theta(\mathbf{a}_0)$.

The goal of the agent is to maximize the expected return at time $t = 0$ (*Equation 1*, reproduced here)

$$J(\theta) = \mathbb{E}_H \left[ \sum_{\tau=0}^{T} \rho_{\tau+1}(H_{0:\tau}) \right], \tag{21}$$

where we have used the time index $\tau$ for notational consistency with the following and made the history-dependence of the rewards explicit. In terms of the probability of each history $H$ occurring, *Equation 19*, we have

$$J(\theta) = \sum_H p_\theta(H) \left[ \sum_{\tau=0}^{T} \rho_{\tau+1}(H_{0:\tau}) \right], \tag{22}$$

where the generic sum over $H$ may include both sums over discrete variables and integrals over continuous variables. Since, for any $\tau = 0, \ldots, T$,

$$p_\theta(H) = p_\theta(H_{0:T}) = p_\theta(H_{\tau+1:T} | H_{0:\tau}) p_\theta(H_{0:\tau}) \tag{23}$$

(cf. *Equation 18*), we can simplify *Equation 22* to

$$J(\theta) = \sum_{\tau=0}^{T} \sum_H p_\theta(H) \rho_{\tau+1}(H_{0:\tau}) \tag{24}$$

$$= \sum_{\tau=0}^{T} \sum_{H_{0:\tau}} p_\theta(H_{0:\tau}) \rho_{\tau+1}(H_{0:\tau}) \sum_{H_{\tau+1:T}} p_\theta(H_{\tau+1:T} | H_{0:\tau}) \tag{25}$$

$$= \sum_{\tau=0}^{T} \sum_{H_{0:\tau}} p_\theta(H_{0:\tau}) \rho_{\tau+1}(H_{0:\tau}). \tag{26}$$

This simplification is used below to formalize the intuition that present actions do not influence past rewards. Using the 'likelihood-ratio trick'

$$\nabla_\theta f(\theta) = f(\theta) \frac{\nabla_\theta f(\theta)}{f(\theta)} = f(\theta) \nabla_\theta \log f(\theta), \tag{27}$$

we can write

$$\nabla_\theta J(\theta) = \sum_{\tau=0}^{T} \sum_{H_{0:\tau}} [\nabla_\theta p_\theta(H_{0:\tau})] \rho_{\tau+1}(H_{0:\tau}) \tag{28}$$

$$= \sum_{\tau=0}^{T} \sum_{H_{0:\tau}} p_\theta(H_{0:\tau}) [\nabla_\theta \log p_\theta(H_{0:\tau})] \rho_{\tau+1}(H_{0:\tau}). \tag{29}$$

From *Equation 20* we therefore have

$$\nabla_\theta J(\theta) = \sum_{\tau=0}^{T} \sum_{H_{0:\tau}} p_\theta(H_{0:\tau}) \left[ \sum_{t=0}^{\tau} \nabla_\theta \log \pi_\theta(\mathbf{a}_t | \mathbf{u}_{1:t}) \right] \rho_{\tau+1}(H_{0:\tau}) \tag{30}$$

$$= \sum_H p_\theta(H) \sum_{\tau=0}^{T} \sum_{t=0}^{\tau} [\nabla_\theta \log \pi_\theta(\mathbf{a}_t | \mathbf{u}_{1:t})] \rho_{\tau+1}(H_{0:\tau}) \tag{31}$$

$$= \sum_H p_\theta(H) \sum_{t=0}^{T} \sum_{\tau=t}^{T} [\nabla_\theta \log \pi_\theta(\mathbf{a}_t | \mathbf{u}_{1:t})] \rho_{\tau+1}(H_{0:\tau}) \tag{32}$$

$$= \sum_H p_\theta(H) \sum_{t=0}^{T} \nabla_\theta \log \pi_\theta(\mathbf{a}_t | \mathbf{u}_{1:t}) \left[ \sum_{\tau=t}^{T} \rho_{\tau+1}(H_{0:\tau}) \right], \tag{33}$$

where we have 'undone' *Equation 23* to recover the sum over the full histories $H$ in going from *Equation 30* to *Equation 31*. We then obtain the first terms of *Equation 2* and *Equation 3* by estimating the sum over all $H$ by $N_{\text{trials}}$ samples from the agent's experience.

In *Equation 22* it is evident that, while subtracting any constant $b$ from the reward $J(\theta)$ will not affect the *gradient* with respect to $\theta$, it can reduce the variance of the stochastic estimate (*Equation 33*) from a finite number of trials. Indeed, it is possible to use this invariance to find an 'optimal' value of the constant baseline that minimizes the variance of the gradient estimate (*Peters and Schaal, 2008*). In practice, however, it is more useful to have a history-dependent baseline that attempts to predict the future return at every time (*Wierstra et al., 2009*; *Mnih et al., 2014*; *Zaremba and Sutskever, 2016*). We therefore introduce a second network, called the *value network*, that uses the selected actions $\mathbf{a}_{1:t}$ and the activity of the decision network $\mathbf{r}_{1:t}^{\pi}$ to predict the future return $\sum_{\tau=t}^{T} \rho_{\tau+1}$ by minimizing the squared error (*Equations 4–5*). Intuitively, such a baseline is appealing because the terms in the gradient of *Equation 3* are nonzero only if the actual return deviates from what was predicted by the value network.

## Discretized network equations and initialization

Carrying out the discretization of *Equations 9–12* in time steps of $\Delta t$, we obtain

$$\lambda_t = \text{sigmoid}(W_{\text{rec}}^{\lambda}\mathbf{r}_{t-1} + W_{\text{in}}^{\lambda}\mathbf{u}_t + \mathbf{b}^{\lambda}), \tag{34}$$

$$\gamma_t = \text{sigmoid}(W_{\text{rec}}^{\gamma}\mathbf{r}_{t-1} + W_{\text{in}}^{\gamma}\mathbf{u}_t + \mathbf{b}^{\gamma}), \tag{35}$$

$$\mathbf{x}_t = (1 - \alpha\lambda_t) \odot \mathbf{x}_{t-1} + \alpha\lambda_t \odot \left[ W_{\text{rec}}(\gamma_t \odot \mathbf{r}_{t-1}) + W_{\text{in}}\mathbf{u}_t + \mathbf{b} + \sqrt{2\alpha^{-1}\sigma_{\text{rec}}^2}\mathbf{N}(0,1) \right], \tag{36}$$

$$\mathbf{r}_t = [\mathbf{x}_t]_+ \tag{37}$$

for $t = 1, \ldots, T$, where $\alpha = \Delta t/\tau$ and $\mathbf{N}(0,1)$ are normally distributed random numbers with zero mean and unit variance. We note that the rectified-linear activation function appears in different positions compared to standard GRUs, which merely reflects the choice of using 'synaptic currents' as the dynamical variable rather than directly using firing rates as the dynamical variable. One small advantage of this choice is that we can train the unconstrained initial conditions $\mathbf{x}_0$ rather than the non-negatively constrained firing rates $\mathbf{r}_0$.

The biases $\mathbf{b}^{\lambda}$, $\mathbf{b}^{\gamma}$, and $\mathbf{b}$, as well as the readout weights $W_{\text{out}}^{\pi}$ and $W_{\text{out}}^{v}$, were initialized to zero. The biases for the policy readout $\mathbf{b}_{\text{out}}^{\pi}$ were initially set to zero, while the value network bias $b_{\text{out}}^{v}$ was initially set to the 'reward' for an aborted trial, $-1$. The entries of the input weight matrices $W_{\text{in}}^{\gamma}$, $W_{\text{in}}^{\lambda}$, and $W_{\text{in}}$ for both decision and value networks were drawn from a zero-mean Gaussian distribution with variance $K/N_{\text{in}}^2$. For the recurrent weight matrices $W_{\text{rec}}$, $W_{\text{rec}}^{\lambda}$, and $W_{\text{rec}}^{\gamma}$, the $K$ nonzero entries in each row were initialized from a gamma distribution $\Gamma(\alpha, \beta)$ with $\alpha = \beta = 4$, with each entry multiplied randomly by $\pm 1$; the entire matrix was then scaled such that the spectral radius—the largest absolute value of the eigenvalues—was exactly $\rho_0$. Although we also successfully trained networks starting from normally distributed weights, we found it convenient to control the sign and magnitude of the weights independently. The initial conditions $\mathbf{x}_0$, which are also trained, were set to 0.5 for all units before the start of training. We implemented the networks in the Python machine learning library Theano (*The Theano Development Team, 2016*).

## Adam SGD with gradient clipping

We used a recently developed version of stochastic gradient descent known as Adam, for *ada*ptive *moment* estimation (*Kingma and Ba, 2015*), together with gradient clipping to prevent exploding gradients (*Graves, 2013*; *Pascanu et al., 2013b*). For clarity, in this section we use vector notation $\boldsymbol{\theta}$ to indicate the set of all parameters being optimized and the subscript $k$ to indicate a specific parameter $\theta_k$. At each iteration $i > 0$, let

$$\mathbf{g}^{(i)} = \frac{\partial \mathcal{L}}{\partial \boldsymbol{\theta}}\bigg|_{\boldsymbol{\theta}=\boldsymbol{\theta}^{(i-1)}} \tag{38}$$

be the gradient of the objective function $\mathcal{L}$ with respect to the parameters $\boldsymbol{\theta}$. We first clip the gradient if its norm $|\mathbf{g}^{(i)}|$ exceeds a maximum $\Gamma$ (see *Table 1*), i.e.,

$$\hat{\mathbf{g}}^{(i)} = \mathbf{g}^{(i)} \times \min\left(1, \frac{\Gamma}{|\mathbf{g}^{(i)}|}\right). \tag{39}$$

Each parameter $\theta_k$ is then updated according to

$$\theta_k^{(i)} = \theta_k^{(i-1)} - \eta \frac{\sqrt{1-\beta_2^i}}{1-\beta_1^i} \frac{m_k^{(i)}}{\sqrt{v_k^{(i)}} + \varepsilon}, \tag{40}$$

where $\eta$ is the base learning rate and the moving averages

$$\mathbf{m}^{(i)} = \beta_1 \mathbf{m}^{(i-1)} + (1-\beta_1)\hat{\mathbf{g}}^{(i)}, \tag{41}$$

$$\mathbf{v}^{(i)} = \beta_2 \mathbf{v}^{(i-1)} + (1-\beta_2)\left[\hat{\mathbf{g}}^{(i)}\right]^2 \tag{42}$$

estimate the first and second (uncentered) moments of the gradient. Initially, $\mathbf{m}^{(0)} = \mathbf{v}^{(0)} = 0$. These moments allow each parameter to be updated in *Equation 40* according to adaptive learning rates, such that parameters whose gradients exhibit high uncertainty and hence small 'signal-to-noise ratio' lead to smaller learning rates.

Except for the base learning rate $\eta$ (see *Table 1*), we used the parameter values suggested in *Kingma and Ba (2015)*:

$$\beta_1 = 0.9,$$
$$\beta_2 = 0.999,$$
$$\varepsilon = 10^{-8}.$$

## Computer code

All code used in this work, including code for generating the figures, is available at http://github.com/xjwanglab/pyrl.

## Acknowledgements

We thank R Pascanu, K Cho, E Ohran, E Russek, and G Wayne for valuable discussions. This work was supported by Office of Naval Research Grant N00014-13-1-0297 and a Google Computational Neuroscience Grant.

## Additional information

### Funding

| Funder | Grant reference number | Author |
| --- | --- | --- |
| Office of Naval Research | N00014-13-1-0297 | H Francis Song<br>Guangyu R Yang<br>Xiao-Jing Wang |
| Google | | H Francis Song<br>Guangyu R Yang<br>Xiao-Jing Wang |

The funders had no role in study design, data collection and interpretation, or the decision to submit the work for publication.

### Author contributions

HFS, Conceptualization, Software, Formal analysis, Writing—original draft, Writing—review and editing; GRY, Formal analysis, Writing—original draft, Writing—review and editing; X-JW, Conceptualization, Formal analysis, Writing—original draft, Writing—review and editing

### Author ORCIDs

Xiao-Jing Wang, http://orcid.org/0000-0003-3124-8474

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
