## [Decision Letter]

Thank you for submitting your article "Reward-based training of recurrent neural networks for cognitive and value-based tasks" for consideration by *eLife*. Your article has been reviewed by three peer reviewers, including Sam Gershman (Reviewer #2) and Mattia Rigotti (Reviewer #3), and the evaluation has been overseen by Timothy Behrens as the Senior and Reviewing editor.

The reviewers have discussed the reviews with one another and the Reviewing Editor has drafted this decision to help you prepare a revised submission.

Summary:

The authors propose to use reward-based training of recurrent neural networks in order to build models able to perform behavioral tasks that are being studied in animal models by the systems neuroscience and neuroelectrophysiology communities. The trained models are then shown to capture features of both the behavior and of electrophysiological data observed in primate experiments. The reward-based training procedure is more general and more realistic (in terms of mimicking the actual primate training) than the supervised training approaches that have been typically employed so far in computational works that compare trained recurrent neural networks to neural recordings. As a result, this training method promises to have a much wider applicability and reach, in terms of the experimental paradigms that can be modeled and the scientific questions that can be asked. The authors then give several examples of the utility of their framework by training models to reproduce a perceptual discrimination task, a context-dependent integration task, a multisensory integration task, a working memory task, and economic choice task.

The paper is well-written and demonstrates an approach to fit recurrent neural networks to neuroscience experiment that is much more general, principled and powerful than the mentioned supervised method typically used so far. Besides being more natural and of wider applicability, the reward-based training technique will also potentially allow investigators to model the evolution of the neural dynamics and behavior as a subject is learning to perform a task, as the authors nicely demonstrate in a handful of landmark primate decision making tasks. These points alone arguably make for a strong paper.

Essential revisions:

Despite the laudable aspects of the work described above, the reviewers unanimously agreed that more needs to be done to demonstrate that the approach will lead to clear new biological knowledge.

All three reviewers raised the issue that it is not clear that the network learning routine is novel from the machine learning perspective, and yet it is completely clear that assumptions made in the model render it only a distant approximation to a biological learning rule. Hence, the reviewers thought that the manuscript needed to be reframed to be clearer what the contribution was. Less emphasis should be placed on the biological realism of the network, and more emphasis should be placed on the biological utility of training a network based on rewards.

Comments from the reviews that address this point are as follows:

I am having problems assessing novelty. If the main novelty is to provide the first neural implementation of the standard "actor-critic" paradigm, then the authors would need to provide an implementation where learning (both in the actor and in the critic) happens through plausible learning rules, which is clearly not the case here (use of backpropagation through time as well as Gated Recurrent Units; the authors acknowledge these limitations in the Discussion). It is not clear that the problem remains learnable (even with the same "teaching signals") under more realistic assumptions. If, on the other hand, the aim was to argue that jointly training a critic facilitates learning of the actor, the authors did not need a neural implementation to make that point, and in fact, it is already well known that policy-gradient methods benefit immensely from variance reduction through fine-grained reward prediction. So, this leaves us with a third alternative message, which is to use these insights + their (non-plausible) neural implementations to try and map the *algorithm* (as opposed to a particular implementation) to specific brain structures/processes by relating model activity to recorded brain activity and behaviour. I think this could have been the main strength of this paper, but is somehow underplayed (in fact, inspection of the activity of the value network is limited to a single page).

The network is quasi-biological in the sense that it is technically a neural network, but the biological constraints on architecture, dynamics and synaptic plasticity are really weak. Should we see this as essentially a phenomenological model, or is there deeper significance to the "neural" aspect? If the latter, then I think a stronger case needs to be made.

My main criticism with the paper is that some presentation choices might mislead some readers into thinking that the application of the policy gradient method to recurrent neural networks trained with backpropagation through time is novel. In fact, the training method used in this paper is essentially the "recurrent policy gradient" algorithm presented among others in one of the papers referenced by the authors (Wierstra et al. 2009). In that paper from the Schmidhuber group the authors utilize LSTMs, instead of GRUs, but otherwise the rest is essentially the same, including the use of a variance reduction method to the estimate of the gradient of the policy function, consisting in a "baseline value function". In addition, in both this paper and in Wierstra et al. 2009 the baseline value function is an RNNs (in the paper under review, the input to the value function is the hidden state of the policy network, but this seems arbitrary).

In fact both ideas – using backpropagation to estimate the policy gradient and using a baseline reinforcement function to reduce the variance of the gradient estimate – are already present in the original Williams 1992 paper (referenced by the authors). The second idea was however probably mostly popularized by a series of papers by Baxter and Bartlett (that apply policy gradients to solving POMDPs), and by the VAPS (Value and Policy Search) algorithm by Baird and More (1999). I feel that the authors should cite these papers. I also suggest that the authors explicitly say that they're using (a modified version of) the "recurrent policy gradient" algorithm presented in the series of papers by Wierstra et al. Since the algorithm already has a name, it might be beneficial to use it. That would also help make the mentioned connections across the literature most transparent.

Whilst there was clear agreement that the biologically-realistic training regime opened up the possibility of asking new biological questions with the network, and that this was exciting, there was also clear agreement that the biological questions and predictions that you have actually made did not lead to the clear new biological insights that the reviewers were expecting. For example, comments from the reviews that spoke to this issue were:

I found the Results section somewhat weak – the Discussion arrived at the point in the Results that I thought was only the end of the warm-up phase. Up to Figure 3, I thought this is all nice but it looks like a sanity check that their architecture does learn what it's supposed to learn, i.e. a sort of reproduction (albeit in an actor-critic architecture) of the results of Song et al. (2016) where the same authors had used stochastic gradient descent to train RNNs on the same family of tasks. It is only in Figure 4 that the authors start inspecting the value network. In light of what I wrote above (namely, knowing that the learning implementation isn't realistic, and that reward prediction is already known to help a lot in policy gradient learning), this was disappointing to me. I was expecting the authors to use their trained value networks to make more specific, experimentally testable predictions about the type of signals you expect to see (and where), the form of synergy predicted between the progress of learning in the task and the quality of reward predictions (by looking at the synergies in the simultaneous training of both nets), etc.

The main empirical observation from the section "Tasks with simple input-output mappings", apart from the fact that the network learns the tasks, is that it exhibits mixed selectivity. This mixed selectivity observation shows up in several of the other sections, but I feel that it is not a particularly strong argument for this model. Many models could produce mixed selectivity, and in any case the prevalence of mixed selectivity is never quantified. Is this really the only empirical constraint from the neural data?

In general, need more explanation for *why* the network reproduces particular empirical phenomena. Which assumptions provide explanatory power? If one were to deviate from these assumptions, would the model no longer reproduce these phenomena? What is *uniquely* explained by this model?

The authors mention that the model allows the Markov property to be relaxed. This seems like an important observation, but there's no demonstration of this in simulations. What empirical phenomena speak to this issue?

The reward baseline idea is interesting, and the authors mention some empirical data possibly consistent with this idea, but they don't report any simulations of lesions, inactivation or pharmacological manipulations to reproduce these effects with their model.

All three reviewers reiterated this point as essential in the Discussion. The basic point is that *eLife* is a biology journal, not a machine learning journal. It needs to be clearer how the new ML advances have led to substantial new biological insight. There was also a clear suggestion in the Discussion that a major advantage of having a network that learns from rewards is the potential to analyse the dynamics of learning itself; the potential to elucidate the limits of task learnability under sparse delayed rewards, and to predict specific patterns of interaction between the learning of the task and the learning of the reward landscape. The reviewers thought that one potential avenue to strengthen the Results section was to focus on these dynamics.

There was also a discussion about deference to the existing literature. You can see several points above that demonstrate concerns along these lines. A further point was also raised:

The argument that actor-critic/policy gradient models are "opposite of the way value functions are usually thought of in neuroscience" (Discussion) seems extreme, since this only applies to value-based model-free algorithms like Q-learning and sarsa. But there is a long tradition of actor-critic models applied to the basal ganglia; see for example Houk et al. (1995), Dayan & Balleine (2002), Joel et al. (2002), O'Doherty et al. (2004), Takahashi et al. (2008), Maia (2010), to name a few.

---

## [Author Response]

*Essential revisions:*

*Despite the laudable aspects of the work described above, the reviewers unanimously agreed that more needs to be done to demonstrate that the approach will lead to clear new biological knowledge.*

*All three reviewers raised the issue that it is not clear that the network learning routine is novel from the machine learning perspective, and yet it is completely clear that assumptions made in the model render it only a distant approximation to a biological learning rule. Hence, the reviewers thought that the manuscript needed to be reframed to be clearer what the contribution was. Less emphasis should be placed on the biological realism of the network, and more emphasis should be placed on the biological utility of training a network based on rewards.*

We hope that the revised manuscript allays some of these concerns, please see below for more detailed responses to the editor and reviewers’ comments.

Here we note two points that were partially raised in the original manuscript and have now been expanded for emphasis and clarification:

We distinguish a learning rule and the neural activity that results from it. The plasticity rule used in this work is not biologically plausible in the sense that it is not one of the known synaptic plasticity rules. While several recent works have made progress toward biologically plausible backpropagation (or doing without BP altogether), we never considered this to be a point of debate. At present there is simply no biologically plausible learning rule capable of training networks for such tasks, in the same manner. What we argue is not that the physiological learning rule is biological, but rather that the resulting circuits, trained in an ethologically relevant manner, operate in a similar way to neural circuits in the brain as demonstrated by neural activity traces that are quite similar to those recorded from behaving animals. This is a nontrivial, even surprising, result, as we can think of no simple reason why this had to be [a point also raised in Zipser & Andersen (1988) for the supervised learning case].

We did not invent the core learning algorithm used in this work, namely recurrent policy gradients with a baseline value network (responsible for computing expected values). Indeed, in some cases we opted not to include state-of-the-art techniques such as Asynchronous Advantage Actor-Critic (Mnih et al., 2016) due to biological *impossibility* (rather than implausibility). What is novel is the application of these techniques, with a few modifications (that are novel), to tasks relevant to systems neuroscience, particularly to the combined study of behavior and electrophysiology for a wide range of tasks. We have therefore reframed the overall paper around its main contributions as laid out at the beginning of the Discussion, to wit: “In this work we have demonstrated reward-based training of recurrent neural networks for both cognitive and value-based tasks. Our main contributions are twofold: first, our work expands the range of tasks and corresponding neural mechanisms that can be studied by analyzing model recurrent neural networks, providing a unified setting in which to study diverse computations and compare to electrophysiological recordings from behaving animals; second, by explicitly incorporating reward into network training, our work makes it possible in the future to more directly address the question of reward-related processes in the brain, for instance the role of value representation that is essential for learning, but not executing, a task.”

*Comments from the reviews that address this point are as follows:*

*I am having problems assessing novelty. If the main novelty is to provide the first neural implementation of the standard "actor-critic" paradigm, then the authors would need to provide an implementation where learning (both in the actor and in the critic) happens through plausible learning rules, which is clearly not the case here (use of backpropagation through time as well as Gated Recurrent Units; the authors acknowledge these limitations in the Discussion). It is not clear that the problem remains learnable (even with the same "teaching signals") under more realistic assumptions. If, on the other hand, the aim was to argue that jointly training a critic facilitates learning of the actor, the authors did not need a neural implementation to make that point, and in fact, it is already well known that policy-gradient methods benefit immensely from variance reduction through fine-grained reward prediction. So, this leaves us with a third alternative message, which is to use these insights + their (non-plausible) neural implementations to try and map the algorithm (as opposed to a particular implementation) to specific brain structures/processes by relating model activity to recorded brain activity and behaviour. I think this could have been the main strength of this paper, but is somehow underplayed (in fact, inspection of the activity of the value network is limited to a single page).*

We agree that this is not the first neural network implementation of the “actor-critic” architecture – many variations of actor-critic are routinely used in machine learning, and as addressed further below, the concept of actor-critic has a rich history in neuroscience, particularly in models of the basal ganglia. We did not intend in any way to imply otherwise, and we have reframed and expanded portions of the manuscript to make this as clear as possible. Again, what is novel is to demonstrate a reinforcement learning framework of training RNNs to perform a number of cognitive and value-based tasks, which is of wide interest to systems neuroscience.

We have added the output of the value network (the expected reward) to Figure 1 to show how the expected return/confidence is computed by performing an “absolute value”-like operation on the accumulated evidence. As we note in an expanded Discussion, however, it is also common in machine learning applications for the policy and value networks to share the same recurrent units, by using a weighted sum of the two losses (reward maximization and reward prediction error) to jointly train the two using a single loss. The question of whether the decision and value networks ought to share the same recurrent network parallels ongoing debate over whether choice and confidence are computed together, or if certain areas such as OFC compute confidence signals locally, and we view this problem as currently unresolved. Computationally, the distinction is expected to be important when there are nonlinear computations required to determine expected return that are not needed to implement the policy.

We were less focused on the *learnability* problem in this situation because there is no question that the tasks can be learned by animals under similar reward conditions, and we assumed that a sufficiently powerful RL algorithm should therefore be able to replicate this learning. We believe that in the future researchers will elucidate the corresponding mechanism in the brain.

*The network is quasi-biological in the sense that it is technically a neural network, but the biological constraints on architecture, dynamics and synaptic plasticity are really weak. Should we see this as essentially a phenomenological model, or is there deeper significance to the "neural" aspect? If the latter, then I think a stronger case needs to be made.*

We agree that the biological constraints are weak, particularly the learning rule, but in our experience it would be a mistake to completely discard the “neural” aspect and view trained RNNs as a purely phenomenological model – after all, the activity of individual units in our trained networks exhibit strikingly similar features to biological neurons recorded in experiments. For instance, it is quite surprising that, after training with a value-based choice task, three major types of units (offer value, chosen value, and choice) found in OFC physiology naturally emerged in our network (Figure 4).

Since no known biologically realistic learning rule exists with the flexibility to learn all the tasks explored in this work, our goal was to use this as a starting point for future development of increasingly biologically realistic models that can still perform the task.

*My main criticism with the paper is that some presentation choices might mislead some readers into thinking that the application of the policy gradient method to recurrent neural networks trained with backpropagation through time is novel. In fact, the training method used in this paper is essentially the "recurrent policy gradient" algorithm presented among others in one of the papers referenced by the authors (Wierstra et al. 2009). In that paper from the Schmidhuber group the authors utilize LSTMs, instead of GRUs, but otherwise the rest is essentially the same, including the use of a variance reduction method to the estimate of the gradient of the policy function, consisting in a "baseline value function". In addition, in both this paper and in Wierstra et al. 2009 the baseline value function is an RNNs (in the paper under review, the input to the value function is the hidden state of the policy network, but this seems arbitrary).*

*In fact both ideas* – *using backpropagation to estimate the policy gradient and using a baseline reinforcement function to reduce the variance of the gradient estimate* – *are already present in the original Williams 1992 paper (referenced by the authors). The second idea was however probably mostly popularized by a series of papers by Baxter and Bartlett (that apply policy gradients to solving POMDPs), and by the VAPS (Value and Policy Search) algorithm by Baird and More (1999). I feel that the authors should cite these papers. I also suggest that the authors explicitly say that they're using (a modified version of) the "recurrent policy gradient" algorithm presented in the series of papers by Wierstra et al. Since the algorithm already has a name, it might be beneficial to use it. That would also help make the mentioned connections across the literature most transparent.*

It was not in any way our intention to imply that this was the first application of policy gradients to RNNs, and it was a big mistake on our part to have taken this to be obvious. We have changed the exposition to make this much clearer, and included the suggested citations. Please note that we were using Peters & Schall (2008) as a sort of review of the GPOMDP method in addition to the Williams derivation. In any case, we now explicitly say that our work is based on Wierstra’s recurrent policy gradient, which again we felt was clear (please note that we used this term explicitly in the Discussion of the original manuscript) but could have been, and has been made, clearer.

We do think, however, that using the hidden state of the policy/decision network as inputs to the value network is 1) not completely arbitrary and 2) is, in fact, novel from the machine learning perspective. It is similar in spirit to jointly training the policy and value networks as one network with a single, combined loss (reward maximization and reward prediction error), while nevertheless treating the two networks separately. Of course, this was not explained well in the manuscript and we have also included additional discussion of this matter. In particular, we now point out how different choices for this architecture parallel ongoing debate over the question of whether decision and confidence are jointly computed in one brain area and read out by another, or if confidence can be computed locally by a different region from signals in the decision area. Within neuroscience this question remains unresolved, and we hope that our approach, and generalizations based on it, can help shed light on the matter.

*Whilst there was clear agreement that the biologically-realistic training regime opened up the possibility of asking new biological questions with the network, and that this was exciting, there was also clear agreement that the biological questions and predictions that you have actually made did not lead to the clear new biological insights that the reviewers were expecting. For example, comments from the reviews that spoke to this issue were:*

*I found the Results section somewhat weak – the Discussion arrived at the point in the Results that I thought was only the end of the warm-up phase. Up to Figure 3, I thought this is all nice but it looks like a sanity check that their architecture does learn what it's supposed to learn, i.e. a sort of reproduction (albeit in an actor-critic architecture) of the results of Song et al. (2016) where the same authors had used stochastic gradient descent to train RNNs on the same family of tasks. It is only in Figure 4 that the authors start inspecting the value network. In light of what I wrote above (namely, knowing that the learning implementation isn't realistic, and that reward prediction is already known to help a lot in policy gradient learning), this was disappointing to me. I was expecting the authors to use their trained value networks to make more specific, experimentally testable predictions about the type of signals you expect to see (and where), the form of synergy predicted between the progress of learning in the task and the quality of reward predictions (by looking at the synergies in the simultaneous training of both nets), etc.*

*The main empirical observation from the section "Tasks with simple input-output mappings", apart from the fact that the network learns the tasks, is that it exhibits mixed selectivity. This mixed selectivity observation shows up in several of the other sections, but I feel that it is not a particularly strong argument for this model. Many models could produce mixed selectivity, and in any case the prevalence of mixed selectivity is never quantified. Is this really the only empirical constraint from the neural data?*

*In general, need more explanation for why the network reproduces particular empirical phenomena. Which assumptions provide explanatory power? If one were to deviate from these assumptions, would the model no longer reproduce these phenomena? What is uniquely explained by this model?*

*The authors mention that the model allows the Markov property to be relaxed. This seems like an important observation, but there's no demonstration of this in simulations. What empirical phenomena speak to this issue?*

The section on “Tasks with simple input-output mappings” was indeed meant as a sanity check, to confirm that any task that was previously trained using supervised learning could also be learned based on reward only. Please note a modified Figure 1 to show the expected return predicted by the value network during the perceptual decision-making task as a function of time, which was previously Figure 1—figure supplement 2. It shows that the expected reward can be computed by an “absolute value”-like operation on the accumulated evidence to compute the expected reward. (It is not simply an absolute value because it also requires a shift, etc.). However, the confidence experiment (Figure 3) is new, not simulated in our previous work using supervised learning.

The tasks considered in this work are in some sense too simple for strong conclusions to be drawn about the role of the value network and we have avoided doing so, although we clearly speculate on possibilities that are of interest for future investigation. Moreover, we found the value network in the economic choice task of some interest precisely because electrophysiologists have recorded from the OFC of monkeys performing a similar task, but it is quite rare that PFC/PPC and OFC are recorded simultaneously so that we had little experimental constraint to work with – a situation that will, we hope, change in the future.

The relaxation of the Markov assumption is more relevant to ongoing work on certain games but we wanted to repeat this point, which was already made in Wierstra et al. (2009). We removed this comment to avoid confusion.

*The reward baseline idea is interesting, and the authors mention some empirical data possibly consistent with this idea, but they don't report any simulations of lesions, inactivation or pharmacological manipulations to reproduce these effects with their model.*

*All three reviewers reiterated this point as essential in the Discussion. The basic point is that eLife is a biology journal, not a machine learning journal. It needs to be clearer how the new ML advances have led to substantial new biological insight. There was also a clear suggestion in the Discussion that a major advantage of having a network that learns from rewards is the potential to analyse the dynamics of learning itself; the potential to elucidate the limits of task learnability under sparse delayed rewards, and to predict specific patterns of interaction between the learning of the task and the learning of the reward landscape. The reviewers thought that one potential avenue to strengthen the Results section was to focus on these dynamics.*

We were very mindful of the fact that *eLife* is a biology journal, and, importantly, that the primary intended audience of our work were neuroscientists. It would be a stretch to say that current research on RL in machine learning can be applied to neuroscience wholesale; we believe, however, that judiciously applying those methods and adapting the parts we can to be even incrementally more biologically plausible is extremely useful, and that was the main goal of our paper. We were less focused on the learnability problem in this situation because there is no question that the tasks can be learned by animals under similar reward conditions, and we assumed that a sufficiently powerful RL algorithm should therefore be able to replicate this learning.

We are very interested in comparing the dynamics of learning in our networks to animals under experimental conditions; however, we simply do not have the necessary data for meaningful comparison. We are working with experimental collaborators to make this a reality. We agree that this is a major advantage of the RL framework.

*There was also a discussion about deference to the existing literature. You can see several points above that demonstrate concerns along these lines. A further point was also raised:*

*The argument that actor-critic/policy gradient models are "opposite of the way value functions are usually thought of in neuroscience" (Discussion) seems extreme, since this only applies to value-based model-free algorithms like Q-learning and sarsa. But there is a long tradition of actor-critic models applied to the basal ganglia; see for example Houk et al. (1995), Dayan & Balleine (2002), Joel et al. (2002), O'Doherty et al. (2004), Takahashi et al. (2008), Maia (2010), to name a few.*

This was at best a sloppy sentence; the sentiment expressed here merely reflects the first author’s own (cortical) bias in interacting with neuroscientists working on reinforcement learning. We are of course aware that actor-critic models have a long and distinguished history in neuroscience, especially in work on basal ganglia. We have added the suggested references along with a sentence in the Discussion acknowledging this history, and removed the problematic phrase in question.